# Spatiotemporal development of expanding bacterial colonies driven by emergent mechanical constraints and nutrient gradients

Harish Kannan [1,12], Hui Sun [2,12] ✉, Mya Warren[3], Tolga Çağlar[3,9], Pantong Yao[4], Brian R. Taylor [3,10], Kinshuk Sahu[5,11], Daotong Ge[1], Matteo Mori[3], David Kleinfeld [3,6], JiaJia Dong [7] ✉, Bo Li [8] ✉ & Terence Hwa [3] ✉

Bacterial colonies growing on solid surfaces can exhibit robust expansion kinetics, with constant radial growth and saturating vertical expansion, suggesting a common developmental program. Here, we study this process for *Escherichia coli* cells using a combination of modeling and experiments. We show that linear radial colony expansion is set by the verticalization of interior cells due to mechanical constraints rather than radial nutrient gradients as commonly assumed. In contrast, vertical expansion slows down from an initial linear regime even while radial expansion continues linearly. This vertical slowdown is due to limitation of cell growth caused by vertical nutrient gradients, exacerbated by concurrent oxygen depletion. Starvation in the colony interior results in a distinct death zone which sets in as vertical expansion slows down, with the death zone increasing in size along with the expanding colony. Thus, our study reveals complex heterogeneity within simple monoclonal bacterial colonies, especially along the vertical dimension. The intricate dynamics of such emergent behavior can be understood quantitatively from an interplay of mechanical constraints and nutrient gradients arising from obligatory metabolic processes.

The formation of bacterial colonies from individual cells on solid surfaces is one of the simplest modes of bacterial growth in nature and in the laboratory. While there is much interest and extensive research in understanding the formation of biofilms by EPS-producing and quorum-sensing cells[1–11], there is a lot to be understood already for simple colonies formed by cells lacking such capabilities and expanding just by the force of their cell growth. Colonies of non-EPS-producing *E. coli* cells growing on hard agar exhibit robust expansion characteristics featuring a constant radial expansion speed, starting a few hours after seeding from a single cell and lasting for several days

[1]Department of Mathematics, University of California, San Diego, La Jolla, CA, USA. [2]Department of Mathematics and Statistics, California State University, Long Beach, CA, USA. [3]Department of Physics, University of California, San Diego, La Jolla, CA, USA. [4]Neurosciences Graduate Program, University of California, San Diego, La Jolla, CA, USA. [5]Department of Bioengineering, University of California, San Diego, La Jolla, CA, USA. [6]Department of Neurobiology, University of California, San Diego, La Jolla, CA, USA. [7]Department of Physics & Astronomy, Bucknell University, Lewisburg, PA, USA. [8]Department of Mathematics and Ph.D. Program in Quantitative Biology, University of California, San Diego, La Jolla, CA, USA. [9]Present address: San Diego Supercomputer Center, University of California, San Diego, La Jolla, CA, USA. [10]Present address: Joint BioEnergy Institute, Emeryville, CA, USA and Systems Biology Department, Sandia National Laboratories, Livermore, CA, USA. [11]Present address: Department of Biomedical Engineering, Duke University, Durham, NC, USA. [12]These authors contributed equally: Harish Kannan, Hui Sun. ✉e-mail: paul.sun@csulb.edu; jiajia.dong@bucknell.edu; bli@ucsd.edu; hwa@ucsd.edu

until nutrient is exhausted[12–15]. On the other hand, the colony height increases linearly with time initially before slowing down and saturating even while the colony radius is still expanding linearly[13,14,16]. The linear radial expansion, observed in colonies of diverse microorganisms[12,15,17,18], has long been attributed to limitation of nutrient penetration from the colony periphery[12,15]. The slowdown in vertical colony expansion has also been recently characterized in quantitative detail for various microorganisms[16]. Several studies, both computational and experimental, have examined the complex metabolic interactions within the colony involving metabolite cross-feeding[13,19–22]. Nevertheless, these studies do not quantitatively connect how metabolism and cell growth affect overall colony expansion and development, a key link towards a bottom-up understanding of the dynamics of colony growth.

One of the earliest models of bacterial colony growth was put forth by Pirt over 50 years ago[12]. Pirt's model described colonies as having a flat pancake-like shape with the colony expanding radially due to growth occurring only at peripheral zones near the leading edge. Further, growth in this peripheral zone was hypothesized to be potentially restricted due to limitation of nutrients such as glucose or oxygen[12,15]. Such a canonical picture of pancake-like colony expansion was first challenged by Wimpenny in 1979[23], by noting that the colony height continuously rises towards the center (unlike a pancake). Wimpenny proposed that overall colony growth is a combination of unrestricted growth at the leading colony edge and nutrient diffusion-limited growth in the interior regions (where Pirt's model predicted lack of growth). However, a mechanistic and quantitative understanding of factors which set the linearity and the rate of colony expansion was lacking until the study by Warren et al. in 2019[24]. In Warren et al., a detailed agent-based computational model including cell-cell and cell-agar mechanical interactions, as well as Monod growth kinetics for a single diffusible carbon source (glucose), was developed to investigate the effect of metabolic and mechanical factors on both radial and vertical expansion speed. Their model predicted that mechanical constraints rather than nutrient shortage limit radial expansion for young colonies during the *establishment phase* (between 10 h and 20 h) of colony development. Additionally, the linear vertical expansion of colony height, driven by the growth of verticalized interior cells, was attributed to limitation of vertical nutrient penetration from agar into the colony. However, Warren et al. did not address the potential consequences of oxygen limitation in the colony interior, which has been observed in colonies beyond the first 24 h[13]. Further, recent experiments found vertical colony expansion to slow down after an initial period[16], in apparent contradiction with the prediction of linear vertical growth by Warren et al.'s model.

In this work, we investigate the growth and maturation of the colony over the course of several days, going well beyond the (aerobic) establishment phase of linear radial and vertical colony growth studied in ref. 24. To understand the causes for the slowdown in colony vertical expansion and to account for essential cellular physiological characteristics, several important metabolic elements are added to the computational model of ref. 24. The key features incorporated in the model include the diffusion of oxygen from the colony boundaries, anaerobic growth of cells in the colony interior, the accompanying excretion and diffusion of fermentation waste (acetate), the requirement of carbon flux for cell maintenance, and eventual cell death if maintenance demand is not met.

Numerical simulations of the expanded model predict that even beyond the establishment phase studied in ref. 24, radial expansion remains limited by mechanical factors rather than nutrient supply at the colony periphery. Additionally, the progressive depletion of the primary carbon source (glucose) in the colony interior is identified as the main cause of vertical slowdown beyond a day of colony development. This glucose depletion, exacerbated by oxygen deprivation in the colony interior, is expected to occur even though the glucose concentration remains high in the agar immediately underneath the colony. These predictions are validated by experiments that track colony growth morphology, including the colony periphery at monolayer resolution, over time under different nutrient conditions. Further, substantial cell death driven by anaerobic carbon starvation in the colony interior is predicted to occur within 1–2 days and the predicted death zone is experimentally validated using two-photon microscopy. Overall, our study highlights the crucial role played by mechanical constraints and emergent nutrient gradients in controlling colony expansion, while also shedding light on starvation induced cell death within the colony, an underexplored aspect of this very common mode of bacterial colony growth.

## Results

### Initial glucose concentration affects vertical but not radial expansion

To probe the potential role of nutrient limitation, in particular, carbon limitation, on colony expansion, we first characterized the morphology of *E. coli* colonies growing on hard agar minimal media plates over the duration of ~2 days, with a range of initial glucose concentration. A non-motile *E. coli* K12 strain harboring constitutive GFP expression (EQ59) was used. Each colony originated from a single cell taken from exponentially growing batch culture in the same growth medium, and the colony seeding density was kept low such that the typical initial inter-colony distance was no smaller than ~1 cm (see Methods).

Measurements of colony dimensions were performed periodically using confocal microscopy (see Methods and Fig. S1a). Colony morphology is radially symmetric after ~10 h (Fig. S1, Supplementary Movie 1, 2, Fig. S2a–h), and the cross-sectional profile, i.e., a plot of the radial dimension at various vertical distance from the agar surface is shown in Fig. 1a–c for three different initial glucose concentrations (10 mM, 20 mM and 30 mM). Colonies grown on plates with lower glucose concentration appear flatter than identically aged colonies grown with higher glucose concentrations, while their radii, defined as the maximum radial dimension (at the agar surface), appear similar (Fig. 1a–c). Further, colony radial expansion was linear in time between 15 h and 50 h for colonies, with similar radial expansion speed till 50 h for the three different glucose concentrations used (Fig. 1d). Vertical colony expansion, defined by the changes in the vertical dimension at the colony center (referred to as "colony height"), had a fast expansion regime between ~5 h and ~15 h (Fig. S2k). Beyond 20 h, vertical expansion slowed down, and colonies grown with higher glucose concentration reach larger heights (Fig. 1e) thereby suggesting that glucose is the limiting nutrient for vertical colony expansion. These observations are also reflected in the overall volume of the colonies (Fig. 1f), estimated from the microscopy images (see Methods). In contrast, reducing the concentration of ammonium, the sole nitrogen source, did not affect either the radial or vertical expansion (Fig. S3) indicating that nitrogen is not the limiting nutrient for colony expansion.

### An agent-based reaction-diffusion model captures colony expansion dynamics

To gain insight into the spatiotemporal dynamics of colony expansion, and particularly the role of metabolism, we expanded a previously introduced agent-based mechanical interaction model[24] by coupling it with reaction-diffusion equations describing various metabolites to study the consequences of emergent gradients of key metabolites affecting bacterial growth and survival. Based on several prior studies on microbial biofilms and colony formation[12,15,19,25–31], oxygen is expected to be quickly depleted in the highly dense colony interior. Thus, it is crucial to include both aerobic and anaerobic glucose metabolism within colony (Fig. 2a, b). Fermentation which occurs to anaerobically growing cells in the absence of alternative electron acceptor necessarily leads to excretion of various fermentative waste

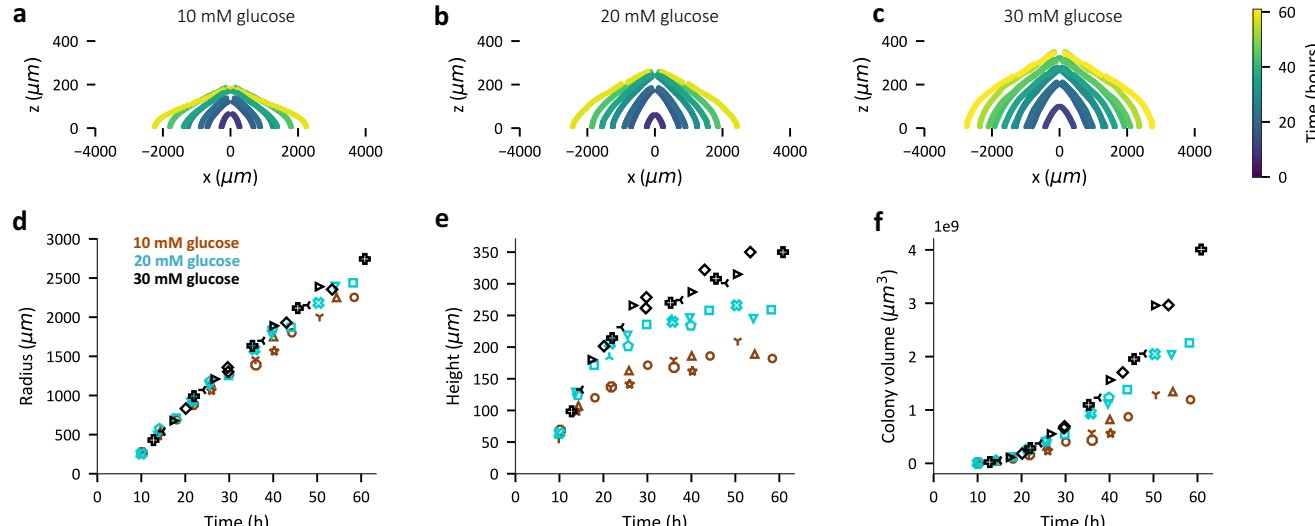

**Fig. 1 | Dependence of *E. coli* colony expansion on initial glucose concentration.** Expansion dynamics of non-motile EQ59 *E. coli* colonies on 1.5 %(w/v) agar plates prepared with a defined concentration of glucose, 10 mM ammonium chloride, and 112 mM phosphate buffer at various times post-inoculation as a single cell. The seeding density of colonies is such that there are ~10 well-separated colony forming units (Fig. S1b) on a petri dish which is 60 mm diameter and has ~8 mm media depth with a total media volume of ~16 ml. The cross-sectional profile of colonies grown on a minimal media hard agar plate with 10 mM glucose (**a**), 20 mM glucose (**b**), and 30 mM glucose (**c**) as carbon source shown for various times (coded by color) post-inoculation. **d**–**f** The radius (µm) (**d**), height (µm) (**e**), and volume (µm³) (**f**) of the colonies plotted against the time (**h**) post-inoculation. Brown, cyan and black represent colonies grown on minimal media plates with 10 mM glucose, 20 mM glucose and 30 mM glucose respectively. In (**d**–**f**) each symbol shape represents an individual biological replicate. The observations reported in this figure on the effect of glucose concentration on colony expansion remains similar for colonies formed by a motile *E. coli* strain (Fig. S15).

products[32]. To reduce model complexity, we kept track of only acetate among all the waste products (Fig. 2b). Importantly, acetate is metabolized in the presence of oxygen (Fig. 2c) while other waste products such as ethanol and formate are not. Additionally, we included the effect of carbon uptake (both glucose and acetate) on cell maintenance[33], the deficit of which, i.e., starvation, leads to cell death[34]. Details of the metabolic model are described in Supplementary Note 2. Most of the parameters used are based on experimental measurements as listed in Supplementary Table 1.

This metabolic model is coupled to other components of the full model (Fig. 2d–f) as follows: 1) A discrete, agent-based model of growth, division, movement, and mechanical interaction of individual bacterial cells with each other and with the agar surface (Fig. 2f); 2) A system of reaction-diffusion partial differential equations (PDEs) capturing the dynamics of the three metabolites (glucose, oxygen, acetate) in both colony and agar. The dynamics of the metabolite concentrations is driven by diffusion, and local consumption/excretion in various regions of the colony based on the metabolic model (Fig. 2e). The reaction-diffusion equations are supplemented with appropriate initial and boundary conditions for the agar and colony region, the latter dynamically defined by the location of the cells (Fig. 2d). Details of the full model are described in the Methods section and in Supplementary Notes 1, 2.

A complete agent-based model in a three-dimensional space becomes computationally expensive due to the sheer number of individual cells (roughly 10⁹ cells) in a mature colony. To keep track of large colonies, in this study we employ a (1 + 1)-dimensional model, with cells occupying a two-dimensional region with one dimension (x) being along the colony-agar interface and the other dimension (z) being perpendicular to it, and with cells allowed to move only in the x-z plane (Supplementary Note 3). With this geometry, the reaction-diffusion equations governing metabolite dynamics are solved in a (1 + 1)-dimensional computational domain (Fig. 2d). Thus, this (1 + 1)-dimensional setting allows the study of both radial and vertical colony expansion while still enabling the simulations to be computationally

tractable. Effects of the reduced spatial dimension in our model are discussed in Supplementary Note 4.

We performed (1 + 1)-dimensional simulations of colony growth on top of a rectangular agar region (Fig. 2d), starting with uniform initial glucose and oxygen concentrations, and no acetate. The width and depth of the agar region are chosen to be comparable to the agar cross-section "available" to a colony in our experiments (Supplementary Table 2). The simulations start with a single cell placed at the center of agar-air interface. The cells elongate and replicate due to growth, pushing each other outward and defining a colony region which would expand radially and vertically. The resulting cross-sectional profiles of such expanding colonies are shown in Fig. 2g–i for three different initial glucose concentrations (10 mM, 20 mM and 30 mM). Similar to experimental observations (Fig. 1a–c), the simulated colonies with lower initial glucose concentrations appear flatter than the colonies with higher glucose concentrations (Fig. 2g–i). The increase of colony width ("radius") is similar for the three cases throughout the duration of the study (~50 h); see Fig. 2j. However, the colony height noticeably increases with higher initial glucose concentrations (Fig. 2k), along with more cells within the colony (Fig. 2l).

### Linear radial colony expansion is not due to nutrient limitation
Consistent with previous reports[4,35], our experiments found cells in the colony interior to be vertically oriented (Fig. S4), while at the colony's radial boundary there is a horizontally oriented monolayer of cells (Fig. S5, Supplementary Movies 3, 4). Since cell growth (which drives cell movement) occurs along the orientation of a cell's long axis, verticalized cells would not be expected to contribute to horizontal colony expansion. Thus the zone of horizontally oriented cells determines the speed of radial expansion, which is proportional to the product of cell growth rate and the size of zone[24].

Our (1 + 1)-dimensional model captures the experimentally observed horizontal orientation of cells at the periphery and the progressive verticalization towards the center of colony (Figs. 3a, S6a, b). Such cell verticalization in the colony interior is due to a mechanical "buckling transition" of horizontally expanding cells[4,24]. In our

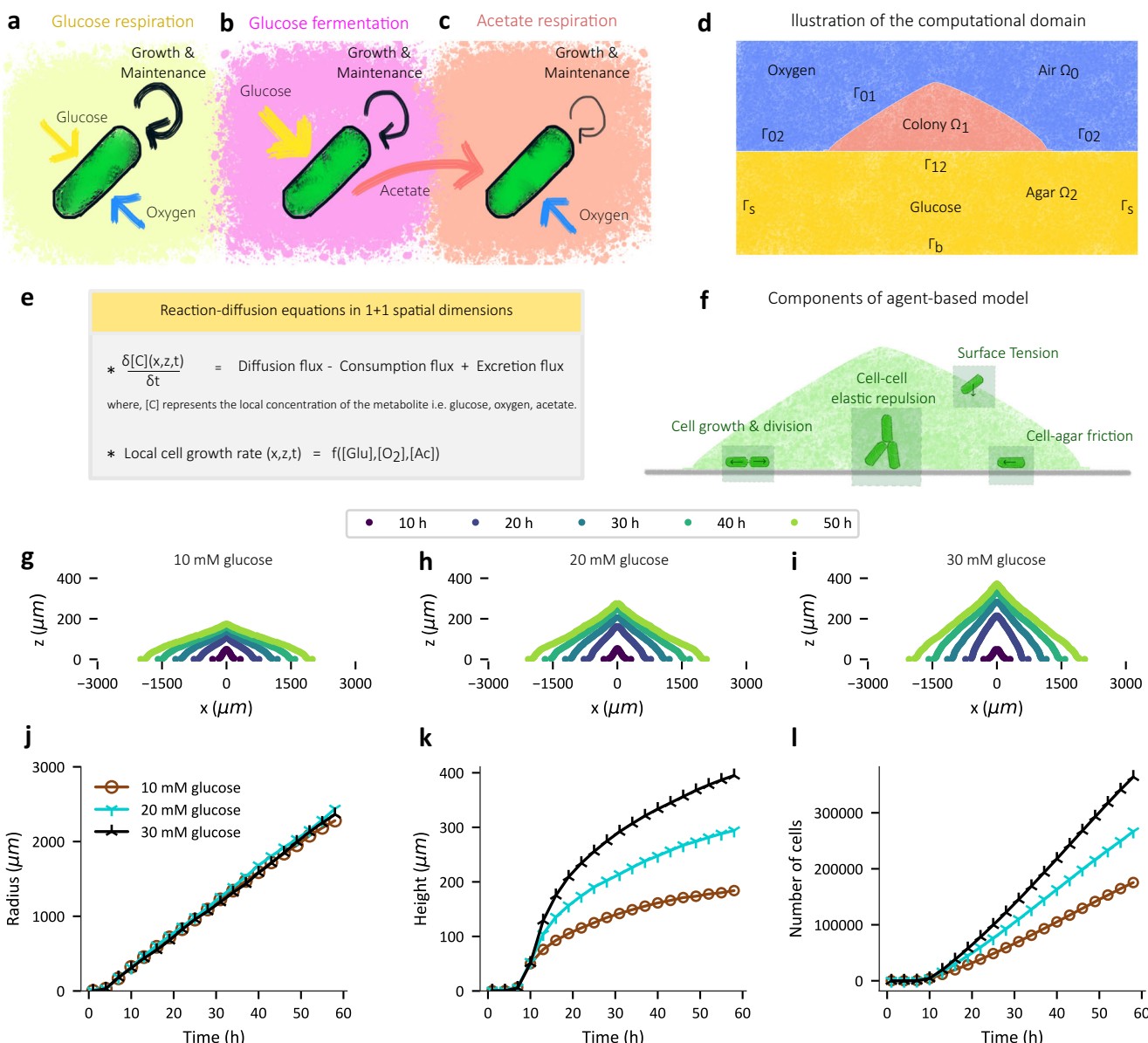

**Fig. 2 | Simulations based on an agent-based and metabolic model capture overall colony expansion dynamics. a–c** The different cell growth and maintenance modes included in our model: aerobic growth and maintenance on glucose (**a**), anaerobic growth and maintenance on glucose (**b**), and aerobic growth and maintenance on acetate (**c**). Anaerobic consumption of glucose by cells is accompanied by acetate excretion indicated by the red arrow in (**b**). Even under aerobic growth on glucose, cells still excrete a small amount of acetate (not indicated) due to overflow metabolism which is also included in our model. **d** Illustration of the computational domain comprising of the colony, agar, and air sub-domains with appropriate boundaries between the sub-domains. **e** A template reaction-diffusion equation used to model the spatiotemporal dynamics of metabolite concentrations. **f** Illustration of agent-based model features involving individual cell growth, division, and movement. The forces experienced by the cells arise from cell-cell, cell-agar interaction as well as surface tension. **g–l** Dynamics of (1 + 1)-dimensional agent-based simulations of colony expansion for different initial glucose concentrations in agar. Simulations start with a single cell on top of an agar region with dimensions of -10 mm×8 mm. The cross-sectional profile of a simulated colony with 10 mM (**g**), 20 mM (**h**), and 30 mM (**i**) initial concentration of glucose in agar, at various times (coded by color) of colony development. **j–l** The radius ($\mu m$) (**j**), height ($\mu m$) (**k**), and number of cells (**l**) in simulated colonies plotted against the time (h) post-inoculation. **j–l** Brown, cyan and black symbols represent simulations with 10 mM, 20 mM and 30 mM initial glucose concentration in agar respectively.

simulations, this orientational configuration remains preserved throughout the duration of simulation (50 h) after the first 10 h (Fig. S6a, b). Consistent with the orientational profile, a non-zero horizontal component of velocity is confined to the peripheral monolayer of the simulated colony (Figs. 3b, c, S6c).

In the simulations by Warren et al.[24], the peripheral monolayer was found to be maintained at a constant width during the duration of their study (the first 20 h). This, coupled with cells growing at maximal rate under glucose replete conditions in the peripheral monolayer,

explains the linearity of radial expansion they reported. Our experimental data shows linear radial expansion to extend well beyond the initial 20 h and even after nutrient depletion (indicated by a slowdown in vertical expansion) (Fig. 1d, e). This may be a reflection of the scenario of Warren et al. (maximum growth rate and constant monolayer width) being applicable beyond the initial establishment phase, or it may result from a nutrient limitation scenario as proposed by Pirt[12], with oxygen being a possible limiting factor[12,15]. To distinguish these two scenarios using our model (which captures linear radial expansion

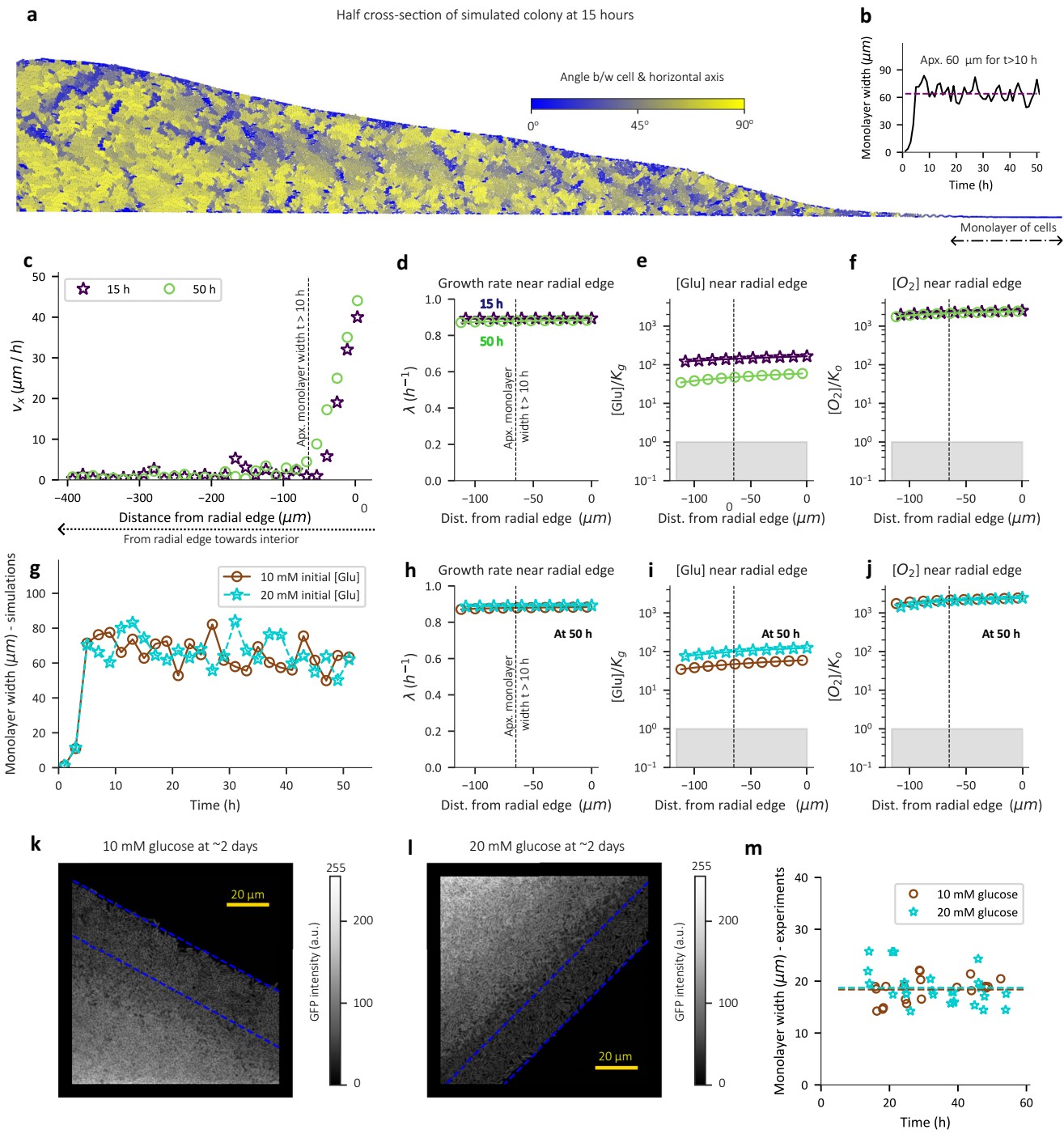

dynamics (Fig. 2j)), we first note that the monolayer width of the simulated colony remains roughly constant between 10 h and 50 h (Fig. 3b). Further, examining the extent of cell growth near the monolayer, cells are seen to grow at close to the maximal rate for at least ~120 μm into the periphery (Fig. 3d), well beyond the ~60 μm monolayer region which determines the radial expansion. This is observed at both 15 h and 50 h of colony development. Crucially, the near maximal growth rate arises because throughout the monolayer, the glucose and oxygen concentration remain well above the values of their respective Monod constants, $K_g$ and $K_o$ (Fig. 3e, f, Supplementary Table 1) even though the interior regions are nutrient limited (Fig. S7).

Thus, in our simulations, linear radial expansion during the first few days of colony development is limited by neither glucose nor oxygen and the linearity of expansion arises due to the constant width

of the radial expansion zone (the monolayer) where cells grow at maximal rate. Further, consistent with the scenario of Warren et al. based on buckling mechanics[24], the monolayer width is not affected by a change in glucose concentration (Fig. 3g). This invariance of the monolayer width explains the independence of radial expansion dynamics of the simulated colony on the initial glucose concentration during the first 50 h (Fig. 2j), since the growth of cells in the monolayer is not limited by nutrients (Fig. 3h–j).

To experimentally test the predicted invariance of monolayer width, we tracked the monolayer width of colonies grown with 10 mM and 20 mM glucose. As predicted, the monolayer width remains roughly constant (~20 μm) throughout the first 50 h and is not affected by the initial glucose concentration (Fig. 3k–m). The quantitative difference in the magnitude of the monolayer width between our

**Fig. 3 | Radial expansion of colony is not limited by nutrients. a** Snapshot of cells within a 15 h old (1 + 1)-dimensional simulated colony with 10 mM initial glucose concentration. Cells are colored based on angle made with the horizontal axis (agar axis) where blue represents horizontal orientation and yellow represents vertical orientation. **b** Monolayer width (μm) of colony (averaged on both peripheral ends of colony) at various times for simulation with 10 mM initial glucose concentration. **c** The horizontal velocity component (μm/h) of cells located at the colony-agar interface in simulated colonies plotted as a function of horizontal distance from the peripheral edge of the colony at 15 h and 50 h. A distance of zero represents the radial edge of the colony. **d**–**f** The growth rate (**d**), glucose concentration (**e**), and oxygen concentration (**f**) at the colony-agar interface is plotted against the distance from the radial edge of the colony at 15 h and 50 h of simulated colony development. The nutrient concentrations in (**e**) and (**f**) are normalized by the respective Monod constants and the grey shaded region represents concentration values less than the respective Monod constant. **c**–**f** are for simulations with 10 mM initial glucose concentration and the dashed vertical line represents the ~60 μm monolayer width for t > 10 h. **g** The monolayer width (μm) of the simulated colonies plotted against time for 10 mM initial concentration in agar (brown) and 20 mM

initial glucose concentration (cyan) in agar. **h**–**j** The growth rate (**h**), concentration of glucose (**i**), and oxygen (**j**) at the colony-agar interface normalized by the respective Monod constant plotted against the distance from the radial edge of the colony after 50 h of simulated colony development. A distance of zero represents the radial edge of the colony. The grey shaded region represents concentrations below the Monod constant. In (**g**–**j**), brown circles represent simulations with 10 mM initial glucose concentration in agar while cyan stars represent simulations with 20 mM initial glucose concentration in agar. **k**, **l** z-projection of GFP fluorescence near the radial edge of a ~2 day old EQ59 colony grown with 10 mM glucose (**k**), and 20 mM glucose (**l**). Blue dashed lines mark the boundaries of the monolayer region identified using the spatial profile of GFP intensity (see methods and Fig. S5 for an example). **m** The width of the monolayer region (μm) at different times post-inoculation (h) for EQ59 colonies grown with 10 mM glucose (circles) and 20 mM glucose agar plates (stars). Each data point represents an individual biological replicate. The value reported for each biological replicate is the monolayer width averaged for multiple ROIs (for most cases *n* > =2: see Source Data) of the colony periphery. The dashed lines in panel m represent the monolayer width value averaged across time for 10 mM (brown) and 20 mM glucose condition (cyan).

simulations (~60 μm) and experiment (~20 μm) likely stem from the (1 + 1)-geometry of simulations which forces the long-axis of the horizontal cells to point radially, thereby leading to an overestimate. Further, the monolayer width in agent-based simulations was also shown to depend on model parameters such as cell-agar friction[24], and thus the model prediction for the magnitude of the monolayer width is at best semi-quantitative.

## Vertical colony expansion is limited by glucose and oxygen

Since cells located in the colony interior are verticalized (experiment - Fig. S4, simulations – Figs. 3a, S6a, b), it is the growth of these verticalized cells that is expected to contribute to vertical expansion. Thus, vertical expansion speed is determined by the thickness of the vertical growth zone and the growth rate of cells located therein[24].

In contrast to horizontal velocity which is highest at the radial edge (Fig. 3c), the vertical component of cell velocity peaks at the center of the simulated colony during early stages of development (blue symbols in Fig. 4a) consistent with the cell orientation profile (Figs. 3a, S6a, b). Over time, the vertical velocity decreases in the colony interior (teal and green symbols, Fig. 4a), mirroring the slowdown in the expansion of the vertical colony height seen in Fig. 2k.

To understand the factors underlying the slowdown in increase of colony height, we thus focus on cells located at the center of the simulated colony (x = 0) at various distances between the colony bottom, i.e., the agar surface (z = 0) and the colony top. Plotting the growth rate (λ) of these cells (Fig. 4b), we find that while almost all cells grow at 10 h (Supplementary Movie 8), the growth at 15 h is restricted to a thin region reaching only ~16 μm above the agar with the cell growth rate sharply dropping away from the agar (teal symbols, Fig. 4b). Further, by 50 h growth vanishes except for cells within 5 μm of the agar surface. Looking further into the distribution of nutrients required for growth, we see that glucose concentration drops below the Monod constant (grey region, Fig. 4c) at vertical distances (~10–20 μm) that are commensurate with the observed drop in growth (Fig. 4b). On the other hand, oxygen concentration drops below the Monod constant within 4 μm away from agar already at 10 h (Fig. 4d), indicating that growth in the simulated colony is largely anaerobic after 10 h (Fig. 4e). A detailed description of the spatiotemporal dynamics of glucose and oxygen concentration throughout the colony and agar is reported in Fig. S7 and Supplementary Movies 5, 6. It is worth noting that the concentration of glucose in the agar underneath the colony is not depleted since glucose consumed is only roughly half of the initial amount even at 50 h of colony development (Fig. S7e) and yet, the local concentration within the colony becomes limiting for growth over time. The overall vertical expansion speed of the colony is proportional to the thickness of the vertical growth zone and the

growth rate of cells located therein[24]. Thus, the progressive decrease in the thickness and cell growth rate in the vertical growth zone (Fig. 4b) explains the gradual slowdown in vertical colony expansion.

Returning to the dependence of vertical colony expansion speed on the initial glucose concentration (Fig. 2k), it is seen that 20 mM initial glucose concentration gives rise to a thicker vertical growth zone (~16 μm at 15 h) compared to that for the 10 mM initial glucose concentration (~10 μm at 15 h) due to increased glucose penetration into the colony (Fig. 4f, g). Thus, the dependence of the thickness of vertical growth zone on the initial glucose concentration (Fig. 4f) rationalizes the effect of initial glucose concentration on vertical colony expansion (Fig. 2k).

Turning to the role of oxygen in colony expansion, the anaerobic consumption of glucose sharply rises at around 10 h of simulated colony development (Fig. 4e). This is a consequence of local oxygen depletion in glucose replete layers near colony-agar interface due to oxygen consumption by the cells there (Figs. 4c, d, i–l, S7, Supplementary Movie 6). The conversion of glucose to biomass is known to be inefficient under anaerobic conditions[32,36] (Supplementary Table 1), resulting in accelerated glucose depletion inside the colony. To experimentally test the impact of oxygen limitation on vertical colony expansion, we tracked the colony dynamics in the presence of an alternate electron acceptor (nitrate[37]), which improves the efficiency of glucose-biomass conversion (Fig. S8b). As expected, nitrate supplementation increases vertical (but not radial) colony expansion for the same glucose concentration (Fig. S8).

## Role of acetate in colony expansion and cell maintenance

Anaerobic consumption of glucose by cells is accompanied by acetate excretion[32,36] (Supplementary Table 1) which can serve as a carbon source for cells located elsewhere provided oxygen is available, but glucose is not (Fig. 2c). We note that even under aerobic growth on glucose, cells still excrete acetate, albeit at a lower level, due to overflow metabolism[38]; this feature is also included in our model (Supplementary Table 1). In our simulations, we see an initial build-up of acetate (<10 mM) within the colony at 15 h (Fig. 5a). With time, acetate concentration decreases towards the top of the colony, where cells depleted of glucose start to consume acetate (Fig. 5a–d, Supplementary Movie 7, 9). This in turn leads to a vertically decreasing concentration of acetate as the consumption of acetate is highest near the colony-air interface, where oxygen is available (Figs. 4kl, 5d).

To understand whether such acetate consumption contributes to cell growth, we first quantified the relationship between cell growth and acetate concentration using batch culture experiments. As seen in Fig. 5e, at the range of acetate concentration (<5 mM) expected within colony after 15 h, growth rate is expected to be below 0.2/h, about 20%

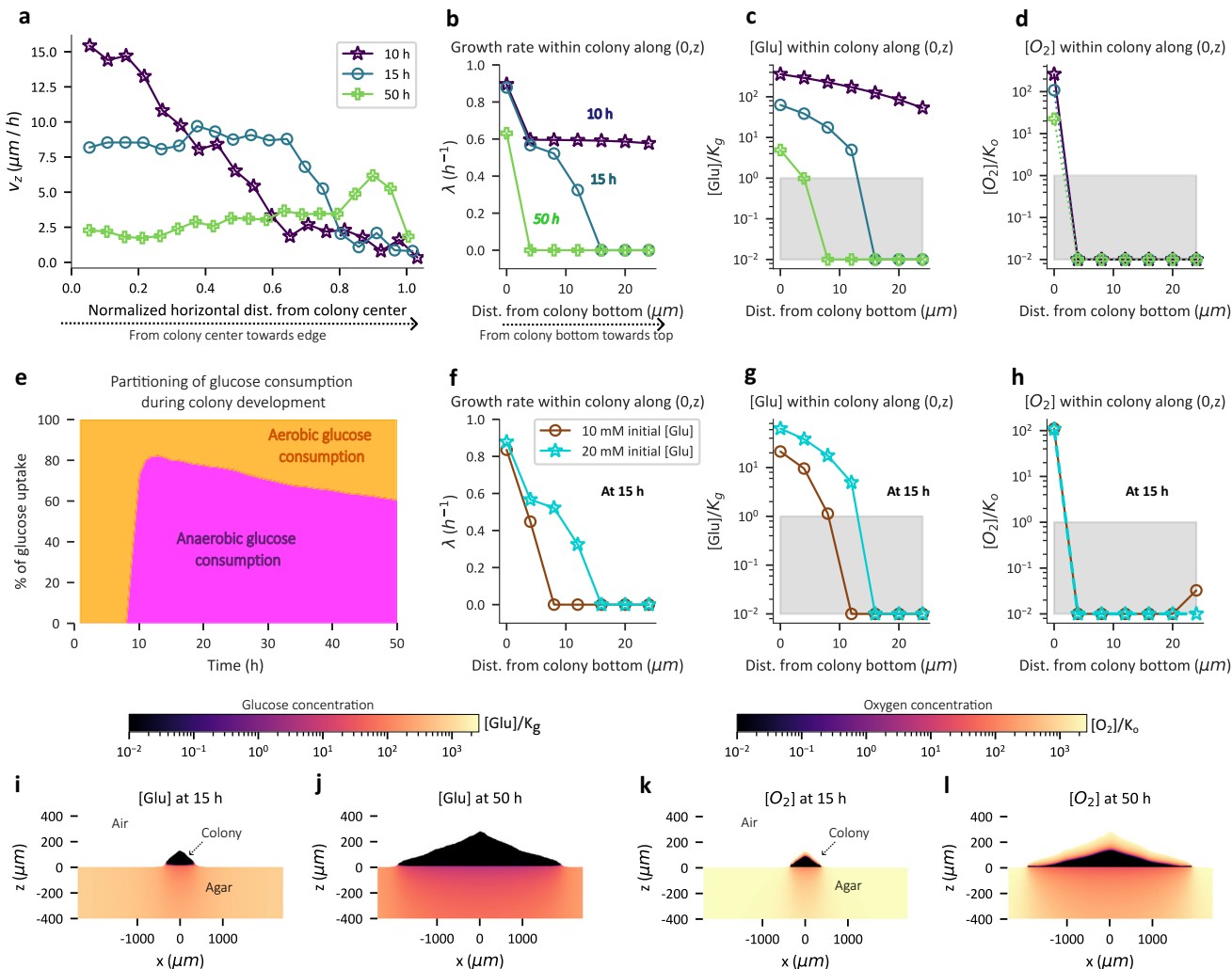

**Fig. 4 | Vertical expansion of colony is limited by glucose. a** The vertical velocity component (μm/h) averaged over the z-coordinate in simulated colonies plotted as a function of horizontal distance from the colony center at 10 h, 15 h and 50 h. A distance of zero represents the colony center. **b–d** The growth rate (**b**), glucose concentration (**c**), and oxygen concentration (**d**) along (0, z) axis of colony is plotted against the distance (z) from colony bottom at 10 h, 15 h and 50 h of simulated colony development. The nutrient concentrations in (**c**) and (**d**) are normalized by the respective Monod constants and the grey shaded region represents concentration values less than the respective Monod constant. **a–d** are for simulations with 20 mM initial glucose concentration. **e** The percentage of glucose uptake flux (integrated over entire colony) towards aerobic (yellow) and anaerobic (magenta) metabolism plotted as a function of time for simulation with 20 mM initial glucose concentration. **f–h** The growth rate (**f**), concentration of glucose (**g**), and oxygen (**h**) plotted along (0,z) axis of colony is plotted against the distance (z) from colony bottom at 15 h of simulated colony development. The grey shaded region represents concentrations below the Monod constant. In (**f–h**), circles represent simulations with 10 mM initial glucose concentration in agar, while stars represent simulations with 20 mM initial glucose concentration in agar. **i–l** The spatial concentration profile of glucose (**i, j**), and oxygen (**k, l**) represented by a colormap, in the colony and the portion of the agar region below the colony for the simulation with 20 mM initial glucose concentration, at 15 h (**i, k**), and 50 h (**j, l**) of colony development.

of the growth rate on glucose, with a sharp dependence on acetate concentration. Indeed, in our simulations, cell growth rate arising from acetate uptake is below 0.2/h at 15 h (Fig. 5f), with virtually no growth due to acetate metabolism by 30 h (Fig. 5g). Instead, at later stages, acetate is being predominantly used for cell maintenance rather than for growth (Fig. 5h–j, Supplementary Movies 9, 10). This arises because when the acetate concentration drops low, acetate uptake for maintenance is prioritized over uptake for growth (explained in Supplementary Note 2).

We note that high levels of acetate accumulation can in principle reduce pH and cause a toxicity within the colony[39–42]. However, in our experiments, we use a high buffer concentration (~112 mM), much exceeding the maximal acetate concentration that can be excreted during growth on glucose (~40 mM acetate even if all of glucose is converted to acetate; see Supplementary Note 5). Even substantially reducing the buffer concentration

(33 mM) has minimal impact on the overall colony expansion dynamics (Fig. S9), indicating that pH-related toxicity is not a concern in our experiments.

## The bulk of the colony is under nutrient starvation
Based on our observations of local glucose depletion in the colony bulk, it comes as no surprise that the proportion of growing cells within the simulated colony sharply drops between 10 to 20 h with the majority of the colony in a non-growing state beyond 20 h (Fig. 6a). This suggests that the sub-exponential increase of colony volume beyond ~20 h in our experiments (Fig. 1f), also captured by our simulations (Fig. 2l), is a consequence of fewer cells within the colony undergoing growth and division. Thus, it is worth noting that this sub-exponential increase of colony volume is an emergent behavior of our model wherein each cell is modeled such that it only responds to its local environment, specifically, the local concentration of nutrients.

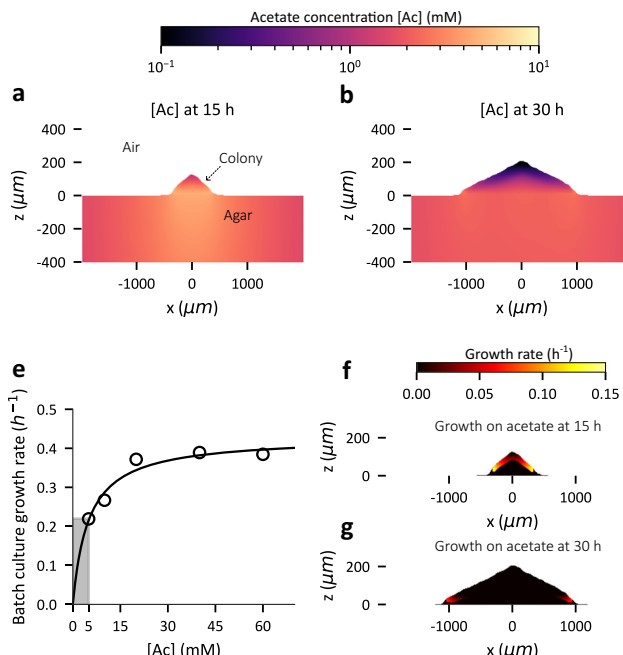

**Fig. 5 | Dynamics of acetate consumption within colony. a–b** The spatial concentration profile of acetate (mM) represented by a colormap, in the colony and the portion of the agar region below the colony for the simulation with 20 mM initial glucose concentration, at 15 h (**a**), and 30 h (**b**) of colony development. **c** Acetate concentration (mM) along the central vertical axis of the simulated colony plotted against the distance (μm) from colony bottom at 10 h, 12 h, 15 h 20 h and 40 h stages of colony development (time coded by color). The grey shaded region represents concentrations below the Monod constant. **d** Glucose (yellow), oxygen (blue), and acetate (salmon) concentration normalized by respective Monod constants $K_g$, $K_o$, and $K_a$ along the central vertical axis of the colony plotted against the vertical distance from colony bottom (μm) at 30 h stage of colony development. The grey shaded region indicates the concentration falling below

the Monod constant. **e** The dependence of EQ59 cell growth rate (h⁻¹) on acetate concentration during batch culture growth with acetate as the sole carbon source (open circles). The solid line represents the fit based on a Lineweaver-Burk plot of data (Fig. S10a, b) and grey shaded region represents the concentration regime below the Monod constant (~5 mM) obtained using the fit. **f, g** Spatial profile of cell growth rate (h⁻¹) arising from acetate metabolism (coded by color) within a 15 h (**f**), and 30 h (**g**) old colony for simulation with 20 mM initial glucose concentration. **h** The percentage of the acetate uptake flux (integrated over entire colony) towards cell growth (light blue) and towards maintenance (green) plotted against time (h) of simulated colony development. **i, j** Spatial distribution of acetate uptake rate for cell maintenance in mmol/gdw/h units (coded by color) within a 15 h (**i**), and 30 h (**j**) old colony for simulation with 20 mM initial glucose concentration.

Given that the majority of the colony is non-growing beyond 20 h (Fig. 6a), we sought to understand the spatiotemporal dynamics of nutrient starvation within the colony. Here we define starvation as a state in which the carbon uptake flux of a cell is less than the maintenance flux (Fig. 6b, see Supplementary Note 2 for further details). Our agent-based model enables us to track the starvation duration for each non-growing cell (Supplementary Note 2). Figure 6c–f and Supplementary Movie 11 provide a visual representation of the spatiotemporal dynamics of starvation within the colony. Virtually no starvation is present at 10 h while after that, a starving central region emerges and expands in size. Upon plotting the starvation duration of cells in the center as a function of colony height (Fig. 6g), we find that it is the interior region of the colony, in-between the agar surface and the colony top surface, that accumulates the longest duration of starvation. This feature is maintained throughout colony development after the onset of starvation, with the duration of starvation increasing as the colony ages (Fig. 6g).

**Starvation induces cell death in the colony interior**

The inevitable physiological consequence of prolonged starvation is cell death[34]. Given that a significant portion of cells in the colony interior are under starvation, we sought to characterize the extent of cell death within the colony. We first quantified the relationship between the duration of starvation and the probability of cell death, by tracking the viability of *E. coli* cells subjected to a controlled duration of carbon starvation using batch culture in aerobic and anaerobic conditions (see Methods). Cells growing exponentially experience a sharp growth arrest (Fig. 7a) upon exhausting the glucose in the medium, thus abruptly entering a phase of glucose starvation, for both

aerobic and anaerobic growth. Figure 7b shows that the viability of culture during such starvation decreases exponentially, at a much faster rate for anaerobic cultures (~2 per day) compared to aerobic cultures (~0.2 per day). From this empirical relationship between the duration of starvation and death, our model is able to predict the fraction of dead cells within a colony based on the duration of starvation and whether the environment is aerobic or anaerobic (Fig. 7c and Supplementary Note 2 for details). The spatiotemporal dynamics of the predicted death zone during the first 50 h of simulated colony development is reported in Supplementary Movie 12. Snapshots reported in Fig. 7d, e show the death zone within a 16-h-old and a 38-h-old simulated colony, respectively, with the death zone predicted to be localized in the interior anoxic core of the colony (Fig. S11a, b). Further, the proportion of dead cells within the colony increases as the colony matures with the onset of death occurring as early as 15 h (Fig. 7f).

To test our model prediction, we used propidium iodide fluorescence along the cross-sectional profile of a colony to identify the *death zone* within a colony (see Methods). Propidium iodide is a fluorescent cell-membrane impermeant dye whose fluorescence intensity increases approximately 20 to 30-fold upon binding to DNA and is routinely used to detect dead cells with compromised cell membranes[34,43]. In order to image fluorescence emitted by cells within the bulk of an intact colony, which is very dense optically, we used two-photon microscopy with high light penetration capabilities[44,45] (See Methods). In Fig. 7g–j, we report the fluorescence from the red emission wavelength band corresponding to propidium iodide fluorescence (colored with magenta) in a colony at 16 h (Fig. 7g, i) and 38 h (Fig. 7h, j) post-inoculation. The corresponding GFP fluorescence which marks the colony boundaries are shown in green. From the

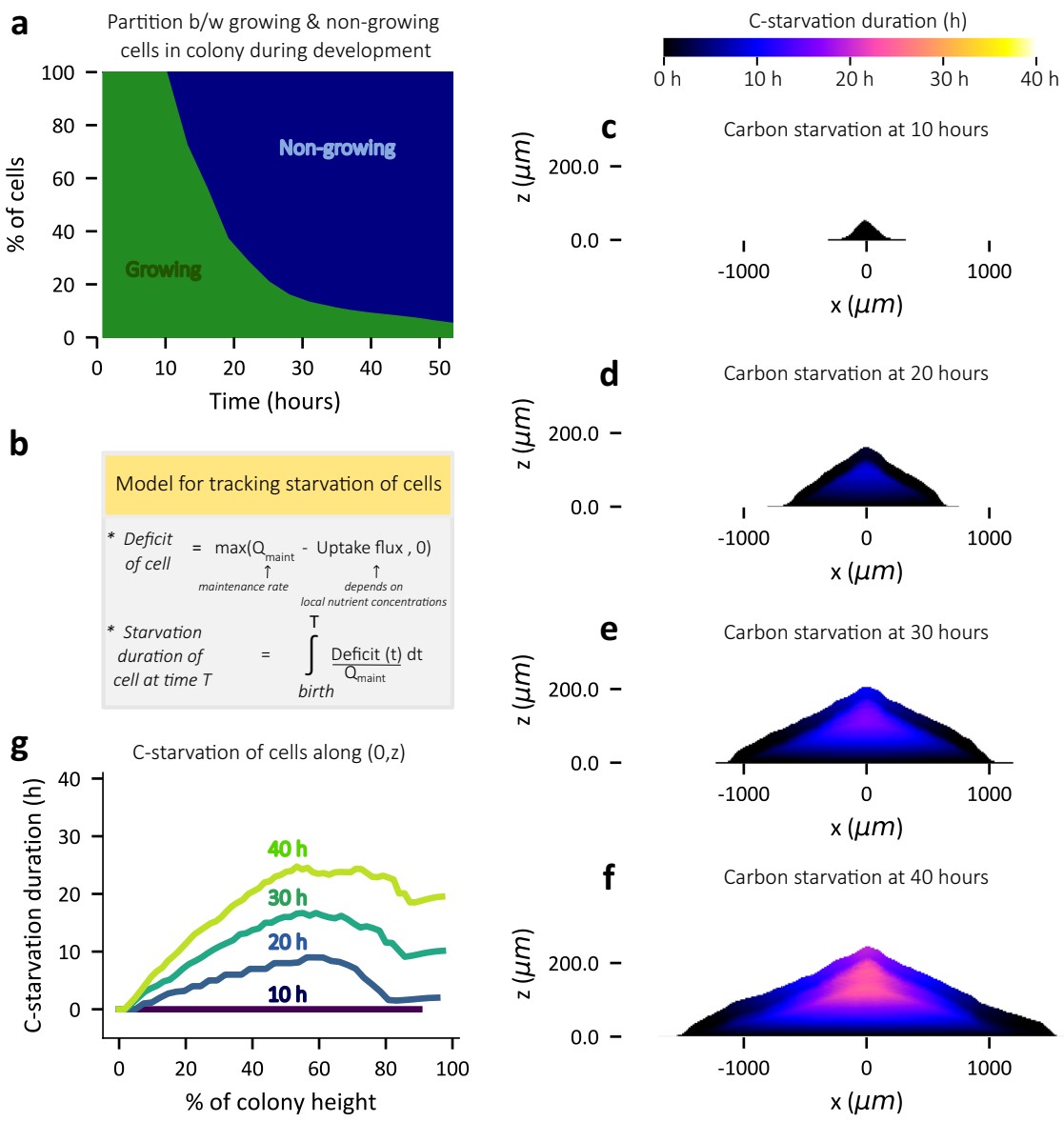

**Fig. 6 | Progressive carbon starvation of cells in the interior during colony development. a** The percentage of growing (green) and non-growing cells (blue) within the colony is plotted against the time of simulated colony development. A cell is classified as growing if its instantaneous growth rate is greater than 0.01 h$^{-1}$ and as non-growing otherwise. **b** A simplified description of our model to track the duration of nutrient starvation for cells within the colony (further details are provided in Supplementary Note 2). The starvation duration of cells (coded by color) within the colony at 10 h (**c**), 20 h (**d**), 30 h (**e**), and 40 h (**f**) of simulated colony development. **g** The starvation duration of cells along the central vertical axis (0,z) of the simulated colony plotted against the vertical position from colony bottom (represented by % of the colony height) at 10 h, 20 h, 30 h and 40 h stages (coded by color) of colony development. Results here are for simulations with 20 mM initial glucose concentration in agar.

projection images shown in Fig. 7g, h, it is seen that the death zone (magenta) is localized to the colony interior away from the radial boundaries. From the vertical cross-section shown in Fig. 7i, j, it is observed that the death zone indicated by the PI fluorescence noticeably increases in size at 38 h (Fig. 7j) in comparison to the zone at 16 h (Fig. 7i). Further, the intensity of propidium iodide fluorescence peaks at around the mid-point of colony height for colonies (Figs. 7i, j, S11g, h, S12–S14), and this is in qualitative agreement with our model prediction of the spatial location of the death zone within the colony (Figs. 7d, e, S11a, b).

## Discussion

In this work, we investigated the development of non-EPS-producing *E. coli* colonies growing on glucose in hard agar. After an initial transient period of exponential growth, the colony expands at a constant radial

speed for at least two days (until eventual glucose exhaustion). On the other hand, the vertical expansion slows down substantially from linear expansion (the establishment phase) beyond the initial 10–20 h. To understand the contribution of mechanical and physiological factors contributing to the radial and vertical expansion, we employ an agent-based computational model which explicitly includes both carbon and oxygen dynamics to account for aerobic and anaerobic metabolism within the colony. Most metabolic model parameters are obtained from direct measurements of batch cultures grown in the corresponding conditions. To accommodate the high computational demand, we adopted a (1 + 1)-dimensional model focusing on a cross-section of the expanding colony. This reduced model is able to capture key dynamical features observed experimentally, including the linear radial expansion, the saturation of vertical expansion, and the death zone in the colony interior. By tracking both the movement and

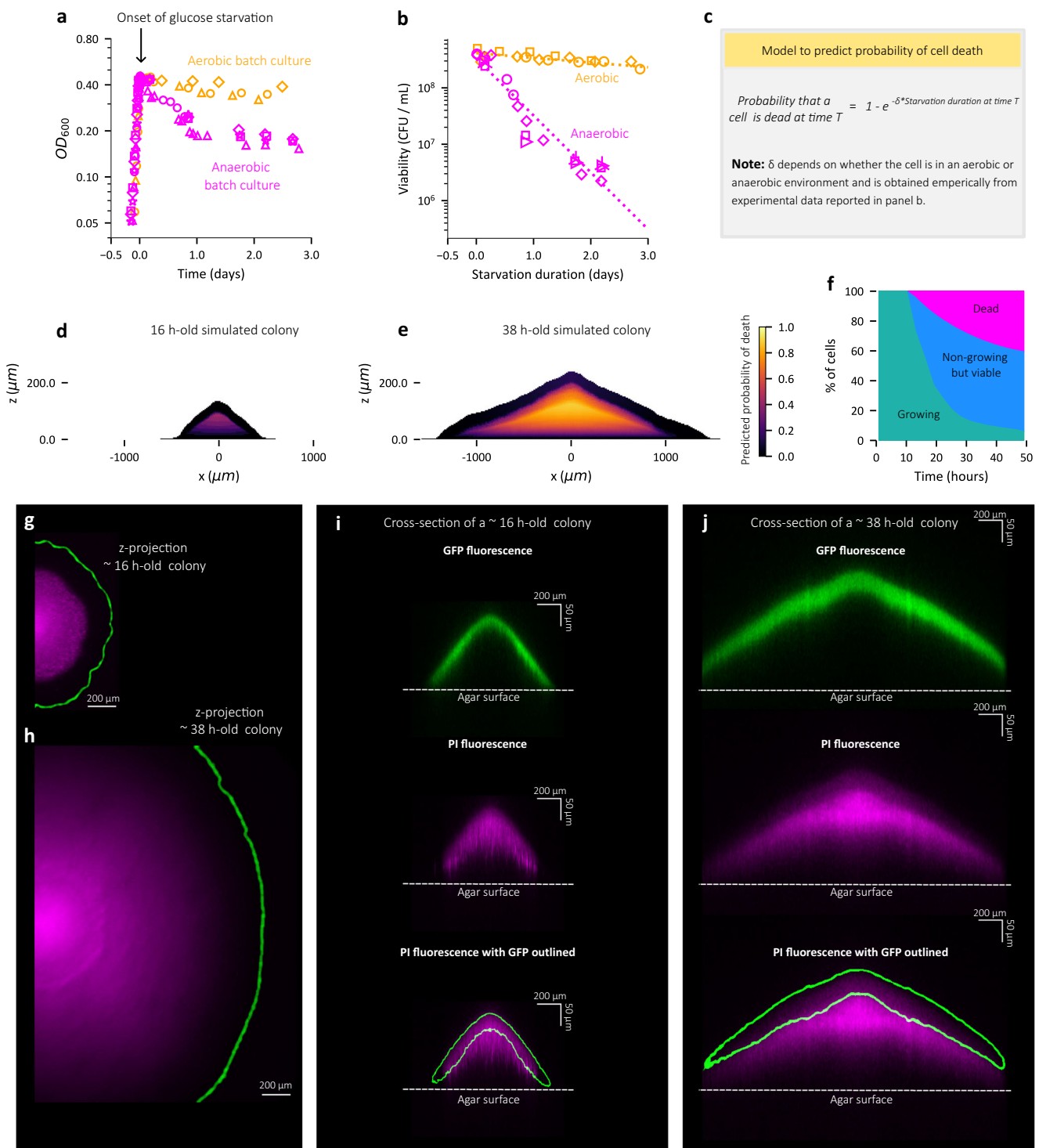

metabolic state of individual cells over time in our simulations, we are able to obtain a detailed spatiotemporal picture of the dynamic progression of growth, starvation and death within the expanding colony.

On radial expansion, our simulation finds both glucose and oxygen to be saturating (i.e., well above their respective $K_c$ in the peripheral zone of the colony, such that neither is limiting for radial expansion. Instead, cell verticalization leading to a constant-width monolayer region (Fig. 3g) is the primary factor driving linear radial expansion in our simulated colony throughout the first 50 h. We tested our model prediction experimentally by changing the glucose concentration. Not only was radial expansion not affected (Fig. 1d), the width of the peripheral monolayer, which determines the speed of

radial expansion, is constant over the 10−50 h period (Fig. 3m) as expected under the cell verticalization scenario. Additionally, we showed that providing an alternative electron acceptor (nitrate) externally, which should alleviate any lack of cell growth due to oxygen limitation, did not affect the linear radial expansion (Fig. S8). These results rule out the hypothesis of nutrient limitation as the cause of linear radial expansion, while they are all consistent with the scenario of mechanical constraints such as cell buckling[24] that result in cell verticalization. This picture on radial expansion is summarized in Fig. 8a.

Our simulations do find the sharp vertical glucose concentration gradient inside the colony to be the primary cause of

**Fig. 7 | Prolonged starvation causes cell death in colony interior within 2 days of development. a** Temporal dynamics of optical density (OD$_{600}$) for aerobic (in yellow) and anaerobic batch cultures (in magenta) of EQ59 cells that are under steady state exponential growth until ~0.4 OD at which point the culture enters growth arrest due to glucose depletion (see Methods). **b** The number of viable cells per mL (colony forming units / mL) of such glucose starved EQ59 cultures is plotted against the duration of glucose starvation (days) for the aerobic cultures (in yellow) and anaerobic cultures (in magenta). By fitting data to a curve of the form $ae^{-\delta t}$, the death rate δ for aerobic and anaerobic glucose starvation are found to be ~0.2 per day and ~2 per day respectively. The yellow and magenta dashed lines represent the fitted curve for aerobic and anaerobic glucose starvation respectively. For (**a, b**), each shape of symbol represents an individual biological replicate for the corresponding culture condition represented by color (yellow - aerobic, magenta – ananerobic) and in (**b**) the value reported for each data point is an average of multiple technical replicates for colony forming units assay (see Methods). **c** Illustration of model to predict the death probability of a cell within the colony based on the duration of starvation introduced in Fig. 6b. **d, e** Model prediction for the probability of cell death (coded by color) within a ~16 h-old simulated colony (**d**), and ~38 h-old simulated colony (**e**). **f** Temporal dynamics of different cell states: growing cells (green), non-growing but viable cells (blue) and dead cells (magenta) within simulated colony. Cells with instantaneous growth rate > 0.01 h$^{-1}$ are counted as growing cells. The number of dead cells within a 4 μm × 4 μm grid inside the colony is estimated as the product of

the number of cells and the average death probability of cells within that grid box. The total number of dead cells is then obtained by summing the dead cell count within all such grids that cover the entire simulated colony. The total number of cells - (number of dead cells + number of growing cells) make up the non-growing but viable population size within the colony. Simulation results presented in (**d–f**) are with 20 mM initial glucose concentration. **g, h** Summed intensity along the vertical axis, i.e. z-projection intensity of propidium iodide fluorescece (colored with magenta LUT) near the radial edge of a EQ59 colony at ~16 h (**g**), and the same colony at ~38 h (**h**) with the edge of the colony determined by GFP fluorescence which is outlined in green. **i, j** GFP fluorescence (in green) and propidium iodide fluorescence (colored with a magenta LUT) in an optical cross-section at the center of a ~16 h old colony (**i**), and the same colony at ~38 h (**j**) obtained using two-photon microscopy (see Methods). The fluorescence from GFP consitutively expressed by cells (top row) is indicated by green outlines in bottom row. In panel (**j**), the radial edge of the colony (shown in panel (**h**)) is beyond the field of view displayed. Data shown in panels (**g–j**) represent the same colony imaged at the specified times after inoculation. Note that GFP requires oxygen to mature[76], and thus, the lack of GFP fluorescence in the colony interior (panels **i, j**) is indicative of oxygen depletion in the colony interior[13]. **g–j** Colony growth was performed on 1.5% agar plates with 20 mM glucose, 10 mM ammonium chloride, 112 mM phosphate buffer and 2.5 μM of propidium iodide and the death zone in biological replicates of EQ59 colonies are shown in Fig. S13.

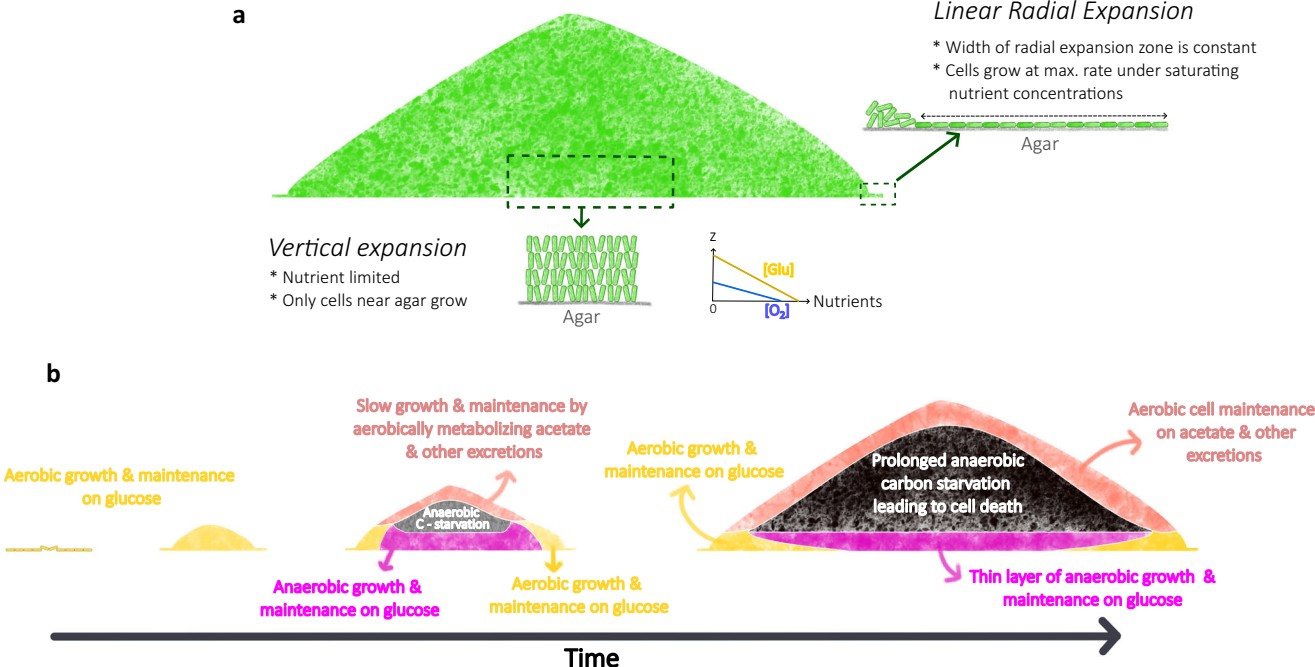

**Fig. 8 | The developmental program of colony expansion dictated by mechanical constraints and emergent nutrient dynamics. a** Overview of the mechanism for colony expansion: Radial expansion is determined by the growth of the peripheral monolayer of horizontally oriented cells. During the linear regime of linear expansion, this peripheral zone of radial expansion remains constant in size. Cells in this region experience a saturating concentration of nutrients (glucose and oxygen) and grow at the maximum rate during the linear regime of radial expansion. Vertical expansion is determined by the growth of vertically oriented interior cells located near the colony-agar interface which is nutrient limited. **b** Overview of the spatiotemporal dynamics of metabolism within colony: Initial stages of colony development involve the transition from a monolayer to a 3D colony during which aerobic glucose metabolism dominates. Later, due to oxygen depletion, cell growth from anaerobic glucose metabolism dominates over cell growth from

aerobic glucose metabolism which is restricted only to the peripheral edges. Due to high consumption flux, glucose is limited to a few μm away from the agar interface. Further, excretions from anaerobic metabolism such as aceate contribute to slow growth and maintenance at the colony-air interface. The colony interior experiences starvation since acetate and other excretions cannot be metabolized without oxygen (oxygen is restricted to a few μm from the colony-air interface). Thus, as time progresses, due to glucose depletion, growth (by aerobic glucose metabolism) is restricted to peripheral edges of the colony with cell maintenance from anaerobic glucose metabolism near agar interface and cell maintenance from aerobic acetate (and other excretions) metabolism at the air interface. Further, due to prolonged anaerobic carbon starvation, a zone of dead cells form in the colony interior.

slowdown in vertical expansion. This severe glucose depletion arises from the high cell density in the colony and is further accelerated by the high glucose consumption flux during anaerobiosis (Supplementary Table 1). In support of the impact of anaerobiosis on vertical expansion, providing an alternate

electron acceptor (nitrate) externally noticeably increases vertical colony expansion (Fig. S8).

In recent years, there has been a renewed interest in understanding the vertical dynamics of biofilm formation on hard substrates[16,46]. A comprehensive experimental study by Bravo et al.[16]

characterized the vertical dynamics of biofilm expansion for various microbial species with nanometer precision measurements of biofilm height. Their findings indicate that the vertical expansion kinetics of monoclonal biofilms formed by diverse microorganisms universally experience a slowdown beyond an initial fast expansion regime. Bravo et al. explained this phenomenon based on a heuristic interface model wherein cell growth was assumed to be restricted to a thin layer at the colony-substrate interface. This assumption is however at odds with findings of other recent studies[19,20,47] which expected the excretion of fermentation products such as acetate also to play important roles at the colony-air interface. This apparent conflict is resolved in our study by recognizing that the contribution of the excreted acetate to colony growth is minor at a quantitative level (Fig. 5, Supplementary Movie 9). Our findings on vertical colony expansion are summarized graphically in Fig. 8a, b.

We regard the most significant finding of this work to be connecting the slowdown in vertical colony expansion to cell death. Our model predicts a localized death zone (Fig. 7d, e) caused by prolonged anaerobic carbon starvation in the colony interior already a day into colony development, as vertical expansion starts to slow down (even while radial expansion is still linear). This prediction is validated by observing propidium iodide fluorescence along the cross-section of the expanding colonies using two-photon microscopy (Figs. 7g–j and S12–S14). In our simulations, we find that acetate metabolism contributes to the carbon flux required for cell maintenance at the colony-air interface (where oxygen can be utilized). Thus, even though acetate contributes minimally to cell growth (Fig. 5f, g and Supplementary Movie 9), it plays a role in keeping the top layers of the colony viable by fueling cell maintenance (Fig. 5i, j and Supplementary Movie 10). Factors governing cell viability/death are summarized in Fig. 8b.

In our metabolic model, acetate is the only fermentation product included due to its large excretion flux[41] and its ability to be utilized in the presence of oxygen (as opposed to e.g., ethanol and formate[32,41]). There are of course also other excretion products that can contribute to cell metabolism close to the colony surface where oxygen is available. In this vein, a recent study by Diaz-Pascual et al.[13] discussed the consumption of alanine by cells at the colony-air interface based on their spatially resolved transcriptomic data for *E. coli* colonies. Alanine can be utilized by cells as both a carbon and nitrogen source. In our experiments, changing the concentration of ammonium did not affect the vertical expansion of colony (Fig. S3), suggesting that nitrogen is not the rate-limiting nutrient for colony expansion. Further, since the excretion of alanine is much lower than acetate during fermentation[36], we expect alanine utilization as a carbon source to be lower than acetate utilization within the colony. Spatial transcriptomics done in Diaz-Pascual et al.[13] did not reveal signatures of acetate cross-feeding by looking at the fold change in expression of genes such as *ackA*, *pta* and *eutD*. At low acetate concentrations (<10 mM), it is likely that the Ack-Pta pathway does not participate in acetate assimilation and instead the Acs pathway is expected to be primarily used by *E. coli* for acetate utilization[48]. Further, genes involved in acetate metabolism (e.g. *acs*, *aceA*, *aceB*, etc.) are part of a larger group of carbon catabolic genes[49] which have an obligatory upregulation during carbon limitation[49–51]. This upregulation of genes involved in acetate metabolism during carbon limitation occurs regardless of whether acetate is utilized or not. Given that the bulk of the colony is expected to be under carbon limitation (Supplementary Movie 11), it would be difficult to detect acetate cross-feeding from fold-change in the expression of genes involved in acetate metabolism. This is because the only cells which are not expected to express acetate metabolism genes are those growing on glucose, which comprise a tiny fraction of the colony even as early as 12 h into development (Supplementary Movie 9).

As seen from Supplementary Movie 10 (bottom row) and Fig. 5, our work predicts acetate utilization to occur primarily near the colony-air surface, which is in agreement with an earlier study by Cole et al.[19]. Further, it should be noted that our observations do not rule out alanine uptake near the colony-air interface. Rather, it is unlikely that cells at the colony-air interface rely primarily on alanine for carbon and nitrogen since acetate (carbon source) and ammonium (nitrogen source) are both expected to be present at higher levels than alanine. Instead, we expect the uptake of alanine at the colony-air interface to play a secondary role, with acetate and ammonium being the primary carbon and nitrogen sources respectively. Also, in this work, we have not included the recycling of lysis product (e.g., amino acids and nucleotides) released by dead cells except indirectly through the adoption of an exponential death rate which is quantitatively influenced by the extent of such nutrient recycling[34,52]. Metabolism derived from a majority of such lysis products (e.g. proteins, amino acids, nucleotides, etc.) requires oxygen for efficient utilization[53] and in principle can fuel cell growth and maintenance at the colony-air interface where oxygen is available. However, the metabolic processes involved are much more complicated and will be the subject of a future study.

What this study has uncovered through a combination of modelling and experiments are basic characteristics of growing bacterial colonies stripped off of the features of EPS-formation and cell differentiation that make the study of biofilms more difficult[2–4,54]. But the findings obtained here on perhaps the simplest non-EPS-producing colonies already reveal a complex pattern of spatiotemporal dynamics. We believe the approaches developed here and the findings obtained represent important steps towards a systemic quantitative understanding of more complex biofilms, including those that involve EPS formation and cell differentiation[54,55]. Indeed, colony-forming microorganisms can exploit the obligatory nutrient gradients formed at different locations and time in developing colonies as spatiotemporal cues to signal the development of more complex spatial structures and temporal growth patterns[11,31,56–60]. Moreover, with substantial cell death occurring in the colony interior, the subsequent release of lysis products provides a fountain of rich metabolic ingredients that can support complex ecological interactions and nurture evolutionary innovations that improve such necromass recycling[52].

## Methods
### Experimental methods
**Bacterial strains.** The strains, EQ54, EQ59 and HE1067 of *E. coli* K12 used in this study were derived from NCM3722[61]. Strains EQ54 and EQ59 were constructed in past studies[24,62]. Briefly, EQ54 was constructed by the deletion of motA gene in NCM3722 to remove cell motility, and EQ59, in addition to a deletion in motA gene harbors constitutive expression of green fluorescent protein (GFP). To construct HE1067, the chromosomal region carrying 'km<sup>r</sup>:*rrnB*T:P*tet-gfp*' present in EQ59 was transferred to the motile strain HE205[63] using P1 transduction[64], thereby yielding HE1067 which is a motile strain that harbors constitutive expression of GFP.

**Colony growth medium and agar plate preparation.** For studying colony growth dynamics, minimal medium agar plates were used. Each agar plate comprised of 1.5% agar (w/v) along with a defined concentration of glucose (carbon source), 10 mM NH₄Cl (nitrogen source), 42 mM NaCl, 0.4 mM MgSO₄.7H₂O, 5.7 mM K₂SO₄ along with a 112 mM phosphate buffer comprising of 77.5 mM K₂HPO₄ and 34.5 mM KH₂PO₄ to maintain a neutral pH (~7.2). For colony growth with 33 mM phosphate buffer (Fig. S9), 23 mM K₂HPO₄, 10 mM KH₂PO₄ and 150 mM NaCl was used with other components of the medium remaining the same. Unless specified, 112 mM buffer medium was used for all experiments. To prepare minimal media agar plates, a 2x concentration of granulated agar (Fisher BioReagents Catalog No. BP9744) was first sterilized by autoclaving (at 121 °C, 15 psi for 20 minutes). Upon cooling down to ~70 °C after autoclaving, the molten 2x

agar solution was mixed with a filter sterilized (0.22 μm filter) solution containing 2x concentration of the remaining media components. The resultant mixture was immediately poured into sterile 60 mm x 15 mm petri dishes (Fisherbrand) and allowed to solidify at room temperature. The total media volume used for each petri-dish was 16 mL. For experiments to image the death zone within colonies, 2.5 μM of propidium iodide was added to the above growth medium during preparation of minimal media agar plates. After preparation, the plates were sealed in plastic sleeves and stored at 4 °C for no longer than one day before use.

**Batch culture growth.** Batch culture growth was performed in a 37 °C water bath shaker operated at 220 rpm. Cells from a glycerol stock stored at −80 °C were streaked on an LB agar plate and incubated overnight at 37 °C. A single colony from this plate was picked and inoculated into LB broth and grown in a shaking water bath for several hours at 37 °C as seed cultures. Seed cultures were then transferred into minimal medium and grown overnight at 37 °C in a shaking water bath as pre-cultures. For batch culture measurements, overnight pre-cultures were diluted to -0.02 $OD_{600}$ in fresh minimal medium and grown at 37 °C in a shaking water bath as experimental cultures. $OD_{600}$ was monitored using a Genesys 20 spectrophotometer (Thermo Scientific). For batch culture growth in minimal medium, the same medium composition as the minimal medium for colony growth, excluding the 1.5% agar was used. For aerobic batch cultures, either 3 mL cultures in 16 mm x 125 mm glass test tubes or 5 mL cultures in 20 mm x 150 mm glass test tubes was used.

For anaerobic experiments, seed culture was done aerobically, while both, overnight precultures and experimental cultures were done anaerobically in Hungate tubes. Same medium composition was used for both aerobic and anaerobic batch cultures. To maintain an anerobic environment for cell growth, Hungate tubes (16 mm × 125 mm) with 7 mL of the medium were shaken at 270 rpm under 7% $CO_2$, 93% $N_2$ atmosphere pressurized to 1.5 atm for 75 min prior to being used for experiments. Cultures were transferred into and out of Hungate tubes with disposable sterile syringes.

**Colony growth in minimal media agar plates.** Seed cultures and pre-cultures were prepared as described above. Overnight pre-cultures were then diluted to -0.02 $OD_{600}$ in fresh minimal medium and grown at 37 °C in a shaking water bath as experimental cultures. When the experimental culture reaches -0.2 OD, an aliquot of the culture was diluted $10^{-6}$ fold in minimal medium. Then -100 μl of the diluted sample was plated on to minimal medium agar plates (dried for -10 minutes in PCR hood and prewarmed to 37 °C prior to use) and spread with sterile glass beads. The dilution factor and plating volume was chosen such that the number of colonies in each plate was -10. The plates were dried under a flame for roughly 10 min after plating, wrapped with parafilm and kept in an inverted position in a 37 °C incubator. At periodic intervals the agar plates were taken out of the incubator and unwrapped for microscopy measurements which last roughly 5 min. Post-microscopy the plates are wrapped with parafilm and placed in an inverted position in a 37 °C incubator again.

**Glucose starvation in batch culture and viability measurements using colony forming units assay.** Seed cultures and pre-cultures were prepared as described above. Overnight pre-cultures were then diluted to -0.02 $OD_{600}$ in fresh minimal medium with a limiting concentration of glucose (2 mM for aerobic cultures and 5.5 mM for anaerobic cultures) such that at -0.4 $OD_{600}$, glucose runs out in the medium and cells in the culture enter growth arrest. The cultures were maintained at 37 °C in a shaking water bath throughout the experiment. To measure the viability of cells in the glucose-starved culture, an aliquot of culture was diluted, and plated on LB agar plates (dried and prewarmed to 37 °C). The liquid sample was then evenly spread on the agar plate with 5–6 sterile glass-beads and the plates were incubated overnight at 37 °C (aerobically). The dilution factor for plating was chosen such that colony forming units (c.f.u) on each plate was roughly between -30-300 c.f.u. Colonies formed after overnight incubation of plates were counted manually, and the count was adjusted based on the corresponding dilution factor to obtain c.f.u. per mL of culture.

**Confocal microscopy.** Colonies growing on agar plates were imaged using a Leica TCS SP8 inverted confocal microscope equipped with a stage housed in a 37 °C temperature-controlled box. The agar plates were uncovered and placed in an inverted position on the stage and a xyz-stack scan was performed to obtain a 3D image of the colony. Fluorescence from GFP that was constitutively expressed by cells was detected using excitation by a 488 nm diode laser and detected with a 10x/0.3 NA air objective and a high sensitivity HyD SP GaAsP detector. A montage of tile-scans was created and stitched together to form a single 3D image using a custom python script.

**Image processing and analysis for colony size measurements.** The cross-sectional profile of a colony was obtained from the stitched z-stack confocal images by computing the bounding radius for each z-coordinate. From the cross-sectional profile, the maximum radial dimension which defines the colony bottom is reported as the radius and the height is computed as the maximal distance from the bottom at which the bounding radius is still non-zero. Further, for estimating the colony volume, the colony was treated as a stack of thin disks i.e., for each z-step the colony is approximated as a disk of thickness corresponding to the z-step (-5 μm) and a radius corresponding to the radial dimension at the particular z-coordinate. Then, the colony volume was estimated as the sum of the volume of these thin disks that make up the colony. The above computations were performed using a custom python script[65–68].

**Two-photon microscopy.** Two-photon laser scanning microscopy with adaptive optics (AO-TPLSM)[45] was employed to image the death zone in colonies. AO-TPLSM was performed with a 10X air objective (0.5 NA; Thorlabs, TL10X-2P). The system aberration was corrected by a deformable mirror (ALPAO, DM97-15), as described in ref. 44, for better image quality. Propidium iodide fluorescence was excited by a tunable femtosecond laser (Coherent, Chameleon Discovery) tuned to 1070 nm, filtered through a red bandpass filter (Semrock, FF01-593/46-25), and detected with a silicon photomultiplier (Hamamatsu, C13366-3050GA). The fluorescence from GFP in the same colony was excited at 930 nm wavelength, filtered through a green bandpass filter (Semrock, FF01-530/55-25), and detected with the silicon photomultiplier. The post-objective power was approximately 50 mW for imaging the propidium iodide fluorescence and 20 mW for GFP fluorescence imaging. Images were analyzed and processed using Fiji[69] and ImageJ[70].

**Imaging of monolayer at colony periphery**
Colonies and the surrounding agar substrate at different stages of development were carefully cut out and transferred to a #1.5 cover glass in an upright position. Another cover glass was gently placed on top of the colony such that the colony and agar are in between two cover glasses. Then, the entire setup was carefully inverted, and the peripheral region of the colony was immediately imaged using a Leica TCS SP8 inverted confocal microscope equipped with a 40x/1.3 NA water objective. GFP fluorescence arising from excitation with a 488 nm diode laser was detected using a HyD GaAsP detector and corresponding brightfield images were acquired using a PMT transmission detector. The width of the monolayer was determined by analyzing the GFP intensity of the z-projection of the peripheral region (which rises sharply at the end of the monolayer region – see Fig. S5) using custom python code[65–68].

## Scanning electron microscopy of colony

The protocol described in ref. 35 was used to perform SEM on colonies. For colony growth to do SEM experiments, minimal agar plates were prepared such that there are two layers of agar by pouring molten agar media in two stages, i.e. ~10 mL was first poured and after solidification the remaining ~6 mL was poured on top, thereby leaving two separate layers. Colony growth dynamics on such plates were similar to growth on plates with standard preparation. After 2 days of incubation, colonies grown on such plates were carefully overlaid with 1.5% agar to preserve their structure. Then the colony along with the surrounding block of agar were then cut out, and the bottom-most layer of the agar was carefully removed, leaving roughly equal agar volume on top and below the colony. The agar blocks containing the colony were transferred onto a filter paper and then placed on a petri dish. The blocks were then incubated in a mixture of 2% EM grade Paraformaldehyde + 2% EM grade glutaraldehyde in 0.1 M Sodium Cacodylate buffer pH7.4 (SC) for 24 h at room temperature to allow fixation to occur. They were washed twice with 4 C 0.1 M SC and then treated with cold 1% (wt/vol) Osmium Tetroxide in 0.1 M SC for 1 h. The samples were then washed twice with distilled water before dehydration in an increasing graded ethanol series (50%, 70%, 90%, 3× 100%, 10 min each). The blocks were then washed twice for 30 min in 100% Hexamethyldisilane (HMDS) before being mounted onto aluminum stubs using double-sided round carbon stickers and placed into a desiccator. After air-drying thoroughly, the blocks were cut in half to visualize the internal structure and orientation of cells within the colony. The samples were then coated in iridium using a Denton Vacuum DESK IV Sputter Coater at the UCSD Nano 3 facility. The samples were visualized using a FEI Quanta 250 (FEI) at the UCSD Nano 3 facility operating at an accelerating voltage of 5 kV under a high vacuum.

## Computational model

Our model is comprised of three components: an agent-based model for individual cells; a metabolic model to determine local cell growth rate; and a system of reaction-diffusion partial differential equations (PDEs) for modeling the spatiotemporal dynamics of metabolism inside the colony. The agent-based model for individual cell activities was constructed in our previous work, Warren et al. 2019[24] and is reviewed in Supplementary Note 1. Here, we present a brief description of the model to determine local cell growth rate and the reaction-diffusion equations for spatiotemporal dynamics of metabolism with additional details provided in Supplementary Note 2. We also provide a brief description of the numerical algorithm used to simulate colony growth with additional details provided in Supplementary Note 3.

**Growth modes and local cell growth rate.** We include three modes of cell growth in our model: aerobic growth on glucose, anaerobic growth on glucose, and aerobic growth on acetate (Fig. 2a–c). The local cell biomass growth rate $\lambda = \lambda(\vec{r}, t)$ at a spatial point $\vec{r}$ within the colony and time $t$, the main quantity that connects the continuum and discrete parts of our hybrid model, is a weighted sum of the growth rates corresponding to the three different growth modes, given by

$$\lambda = \lambda_1\left(\lambda_{g,aer}, C_g\right)\theta_o + \lambda_2\left(\lambda_{g,ana}, C_g\right)\left(1 - \theta_o\right) + \lambda_3(\lambda_{a,aer}, C_a)\left(1 - \theta_g\right)\theta_o,$$

(1)

where $\lambda_1$, $\lambda_2$ and $\lambda_3$ represent the local growth rates for aerobic growth on glucose, anaerobic growth on glucose, and aerobic growth on acetate, respectively. These local growth rates depend on the local concentrations of glucose $C_g = C_g(\vec{r}, t)$, oxygen $C_o = C_o(\vec{r}, t)$, and acetate $C_a = C_a(\vec{r}, t)$, and also on the maximum growth rates, $\lambda_{g,aer}$, $\lambda_{g,ana}$, and $\lambda_{a,aer}$, corresponding to each of the three growth modes, respectively. Further, $\theta_g$, $\theta_o$, $\theta_a$ represent Monod kinetic forms,

namely,

$$\theta_g = \frac{C_g}{C_g + K_g}, \quad \theta_o = \frac{C_o}{C_o + K_o}, \quad \theta_a = \frac{C_a}{C_a + K_a},$$

where $K_g$, $K_o$, and $K_a$ are the Monod constants for glucose, oxygen, and acetate, respectively. In Eq. 1, the factors $\theta_o$ and $(1 - \theta_o)$ serve as weights to decide whether cells are under aerobic conditions (i.e., $\theta_o = 1$) or anaerobic conditions (i.e., $\theta_o = 0$ which implies $1 - \theta_o = 1$). Further, under aerobic conditions, the factor $(1 - \theta_g)$ is used to determine whether cells grow on glucose or acetate, i.e., to model the hierarchical preference of glucose by *E. coli* for growth. In other words, if glucose concentration is saturating, then $\theta_g = 1$ and $1 - \theta_g = 0$. In a model which does not account for cell maintenance, the local rates $\lambda_1$, $\lambda_2$ and $\lambda_3$ for different growth modes would traditionally be defined using Monod kinetic forms.

$$\lambda_1\left(\lambda_{g,aer}, C_g\right) = \lambda_{g,aer}\theta_g, \quad \lambda_2\left(\lambda_{g,ana}, C_g\right)$$
$$= \lambda_{g,ana}\theta_g, \quad \lambda_3(\lambda_{a,aer}, C_a) = \lambda_{a,aer}\theta_a.$$

(2)

However, to incorporate the effects of cell maintenance on the local cell growth rate $\lambda$ (Eq. 1) we appropriately modify the local growth rates $\lambda_1(\lambda_{g,aer}, C_g)$, $\lambda_2(\lambda_{g,ana}, C_g)$, and $\lambda_3(\lambda_{a,aer}, C_a)$ for cell maintenance which is described in Supplementary Note 2. For instance, considering aerobic growth on glucose, we introduce a threshold value $C^*_{g,a}$ for glucose concentration $C_g$ and modify the corresponding growth rate $\lambda_1(\lambda_{g,aer}, C_g)$ to be 0 if $C_g \leq C^*_{g,a}$ and if $C_g > C^*_{g,a}$ we use an appropriately shifted version of the Monod curve to determine the growth rate. The growth rates corresponding to the other two modes, $\lambda_2(\lambda_{g,ana}, C_g)$ and $\lambda_3(\lambda_{a,aer}, C_a)$ are similarly modified. A detailed description of the determination of the threshold concentrations, as well as the explicit formulas for the local cell growth rates are provided in Supplementary Note 2.

**A continuum model for spatiotemporal dynamics of metabolite gradients.** In our simulations, a defined initial concentration of glucose is set in the agar region. Glucose diffuses within the agar and into the colony region where it is taken up by cells for either growth or maintenance in the presence or absence of oxygen. During growth on glucose, cells excrete acetate which together with oxygen can also be taken up by cells for growth and maintenance. We describe these metabolic processes with a coarse-grained model through a set of reaction-diffusion PDEs together with appropriate initial and boundary conditions for the local concentrations $C_g$, $C_o$, and $C_a$ of glucose, oxygen, and acetate respectively. Referring to Fig. 2d, e, the general form of the reaction-diffusion equations modeling the spatiotemporal dynamics of the metabolite concentrations $C = C_g$, $C_o$, or $C_a$ is given by,

$$\partial_t C = D_+ \Delta C + \rho P - \rho Q \quad \text{(in colony region } \Omega_1\text{)},$$

$$\partial_t C = D_- \Delta C \quad \text{(in agar region } \Omega_2\text{)},$$

$$C_- = C_+ \text{ and } D_- \partial_n C_- = D_+ \partial_n C_+ \quad \text{(on the agar-colony interface } \Gamma_{12}\text{)}.$$

Here, $D_-$ and $D_+$ are the diffusion coefficients for the metabolite with concentration $C$ in the agar and colony regions $\Omega_1$ and $\Omega_2$, respectively, $C_-$ and $C_+$ represent the concentration at the interface on the agar domain and colony domain respectively, and $\partial_n$ denotes the normal derivative in the direction from agar to colony. If $C = C_g$, $C_o$, or $C_a$, then we denote the diffusion coefficients by $D_{g,-}$, $D_{g,+}$, $D_{0,-}$, $D_{0,+}$, or $D_{a,-}$, $D_{a,+}$, respectively. In the reaction term $\rho P - \rho Q, \rho$ is the local cell density and, $P$ and $Q$ are the excretion and uptake rates, respectively, of the metabolites within the colony. We model these

rates for glucose $C_g$, oxygen $C_o$ and acetate $C_a$ as follows,

$$P_g = 0 \text{ and } Q_g = q_{g,aer}\lambda_1\theta_o + q_{g,ana}\lambda_2(1-\theta_o) + Q_{g,maint},$$

$$P_o = 0 \text{ and } Q_o = q_{o,g}\lambda_1\theta_o + q_{o,a}\lambda_3(1-\theta_g)\theta_o + Q_{o,maint},$$

$$P_a = p_{a,aer}\lambda_1\theta_o + p_{a,ana}\lambda_2(1-\theta_o) + P_{a,maint},$$

$$Q_a = q_{a,aer}\lambda_3(1-\theta_g)\theta_o + Q_{a,maint}.$$

Here $\lambda_1 = \lambda_1(\lambda_{g,aer}, C_g)$, $\lambda_2 = \lambda_2(\lambda_{g,ana}, C_g)$ and $\lambda_3 = \lambda_3(\lambda_{a,aer}, C_a)$ are given in Eq. 2. The subscripted $p$ and q are constant parameters: $q_{g,aer}$ and $q_{g,ana}$ are the specific uptake rates for glucose during the aerobic and anaerobic growth, respectively; $q_{o,g}$ and $q_{o,a}$ are the specific uptake rates for oxygen during the aerobic growth on glucose and on acetate, respectively; $p_{a,aer}$ and $p_{a,ana}$ are the specific excretion rates for acetate during aerobic and anaerobic growth on glucose, respectively; and $q_{a,aer}$ is the specific uptake rate for acetate during aerobic growth on acetate. Further, $Q_{g,maint}$, $Q_{o,maint}$ and $Q_{a,maint}$ correspondingly represent the uptake rate of glucose, oxygen, and acetate for maintenance, while $P_{a,maint}$ represents the excretion rate of acetate arising from cell maintenance. Explicit mathematical expressions for the maintenance uptake and excretion rates of metabolites are provided in Supplementary Note 2 and the parameter values are listed in Supplementary Table 1.

Flux-free boundary conditions are imposed for the concentrations of glucose $C_g$ and acetate $C_a$ on the boundaries surrounding the colony and agar regions, i.e., $\partial_n C_g = 0$ and $\partial_n C_a = 0$ on $\Gamma_{01} \cup \Gamma_{02} \cup \Gamma_s \cup \Gamma_b$, (Fig. 2d). The boundary conditions for the concentration of oxygen are $\partial_n C_o = 0$ on $\Gamma_s \cup \Gamma_b$ and $C_o = C_{o,0}$ on $\Gamma_{01} \cup \Gamma_{02}$, where $C_{o,0}$ is a constant oxygen concentration. The initial glucose concentration is set to be a constant in the agar region, whereas the initial acetate concentration in agar is set to be 0.

**Numerical algorithm.** Our simulations of colony growth are done iteratively with each iteration consisting of the following main steps, Step 1: Given the spatial coordinates of all the individual cells within the colony, we determine a coarse-grained, smoothened colony boundary. Step 2: With the colony boundary defined from the previous step, we solve the system of reaction-diffusion equations with appropriate boundary conditions in both the agar and colony regions to update the concentrations of glucose, oxygen, and acetate. Step 3: Given the local concentration of the metabolites from the previous step, we update the local growth rate for cells in the colony region. Step 4: Using the local cell growth rates computed in the previous step, we simulate the growth, division, and movement of all the individual cells within the colony with another nested loop of iteration with a smaller time step. Details of the boundary conditions, numerical methods for solving the set of reaction-diffusion equations[71,72] and the numerical schemes[73] used to simulate the velocity and positions of individual cells can be found in Supplementary Note 3. The simulated code was implemented using C++ programming language[74] and run on a High Performance Computing (HPC) cluster.

**Reporting summary**
Further information on research design is available in the Nature Portfolio Reporting Summary linked to this article.

## Data availability
Quantitative data underlying graphs in the figures are provided as source data with this paper. Raw microscopy images which were analyzed to generate quantitative data are available upon request from the corresponding author. Source data are provided with this paper.

## Code availability
Custom computer software and code that support the findings of this study have been deposited to a publicly accessible online repository on GitHub: https://github.com/srjr-hkannan/BacColonyGrowth/[75].

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

## Acknowledgements

We thank Guillaume Castillon from the UCSD - Cellular and Molecular Medicine Electron Microscopy Core, RRID:SCR_022039, for SEM sample preparation and SEM imaging, the UCSD Nano 3 facility and personnel, in particular Jeff Wu and Zhicheng Long for access to the Sputter Coater and the SEM, as well as training and recommendation for imaging. We are grateful to Hannah Dayton from the Perrimon Lab, Harvard Medical School and Lars Dietrich from Columbia University for advice regarding colony preparation for SEM. We also thank Xiongfei Fu and members of the Hwa lab for helpful discussions. This work was supported in part by NSF through grants MCB-2029574 (T.H. and B.L.), MCB-2029580 (H.S.), MCB-2029480 (J.D.), DMR-1702321 (J.D.), PHY153264 (D.K.), DMS-2208465 (B.L.) and by NIH through grant U24 EB028942 (D.K.).

## Author contributions

M.W. and T.H. conceived the experimental study and M.W. performed initial investigations. H.K., M.W., T.C., and T.H. designed experiments and H.K., T.C., K.S., and B.R.T. performed experiments; H.S., M.W., and B.L. designed the numerical study and H.S. developed the parallelized simulation software; H.K., H.S. performed simulations with contributions from D.G. under the supervision of B.L.; H.K., M.W., M.M., and B.R.T. contributed to modelling with supervision from B.L., and T.H. P.Y. performed two-photon microscopy imaging with the supervision of D.K.; T.C. designed image analysis pipeline for colony radius and height measurements, and H.K. performed image analysis. H.K., J.D., B.L. and T.H. wrote the manuscript with input from all the authors.

## Competing interests

The authors declare no competing interests.
