## [Transparent Peer Review file · Nature Communications]

Spatiotemporal development of expanding bacterial colonies driven by emergent mechanical constraints and nutrient gradients

Corresponding Author: Professor Terence Hwa

Version 0:

Reviewer comments:

Reviewer #1

(Remarks to the Author)

In this manuscript, Kannan et al. used a combination of agent-based simulations and time-lapse imaging to paint a comprehensive picture of how bacterial colony grows on hard agar substrate. Colony growth is an old topic; however, the authors have presented convincing evidence that the mechanism underlying the colony growth dynamics differs from what researchers commonly believe in the field. While it is commonly thought that a narrow growth zone at the colony edge arises due to nutrient limitations and causes linear colony expansion, the author proposed, instead, that it is mechanical in origin. They further demonstrated that inside the growing colony, oxygen and glucose limitation leads to complex spatiotemporal dynamics. Overall, the ideas in the manuscript are interesting, and the simulation model is comprehensive. The paper could be further strengthened by a few additional experiments that test their conclusions, particularly those regarding the key simulation parameters underlying particular trends. To sum up, I recommend publication of the manuscript after addressing following concerns/questions.

Major

- 1) Mechanical verticalization was suggested to lead to linear colony expansion in *E. coli*, which is a rod-shaped bacterium. But what about spherical cells that cannot undergo verticalization? If literature evidence suggests that spherical cells also form colony with linearly growing radius, it will be directly contrasting the authors' hypothesis. Can the authors perform similar experimental quantifications with spherical cells such as *Staphylococcus aureus* or *Staphylococcus epidermidis*?
- 2) The metabolic stratification phenomenon is really intriguing. Can the author change the oxygen conditions to test their predictions? One of potential experiments is to include other terminal electron acceptors such as nitrate in the agar plate. In this case, the electron acceptors (NO₃ instead of O₂) come from the agar plate instead of from above, so the growth dynamics should be dramatically altered. See for example experiments in PMID: 23292774.
- 3) Is the colony height calibrated in the experiment? A 10x air objective was used and due to the refractive index mismatch between air and water/biofilm/agar, the information in z could be significantly distorted. This effect is exacerbated as one goes deeper into the sample.
- 4) The experiments and simulations in the paper seem to have been performed only with two initial glucose concentrations (10 mM and 20 mM). Adding, say, one or two more glucose concentrations and confirming that (a) the simulations recapitulate the colony growth kinetics, as in Figs. 1A-B vs. 3A-B, and (b) the two main results in the paper still hold would significantly strengthen their persuasiveness of the paper.
- 5) In the SI, the agent-based simulations and reaction-diffusion equations are presented in three dimensions, before the (1+1)-dimensional approximation is presented as an implementational detail in Section 3.1. But Section 3.1 only discusses the agent-based simulations, not the reaction-diffusion equations, and there is no apparent reference to dimensionality in Section 3.3, which concerns the solution of the reaction-diffusion equations. So in what dimensions were the reaction-diffusion equations solved? Were they solved in 2D, meaning that the nutrients could only diffuse in the radial and vertical directions? Or were they solved in the full 3D domain, and a vertical slice of the nutrient concentrations was taken to be available to the cells being captured by the agent-based simulations?
- 6) Related to the question above, the (1+1)-dimensional approximation is adequately justified, but what are the potential implications of limiting the simulations (and particularly the nutrient diffusion) to two dimensions? Assuming that all the cells

are roughly expanding radially outward, it is perhaps reasonable to limit cell movement within a vertical cross-section of the colony to only the radial and vertical directions, but this is not the case for the nutrients. How might adding a degree of freedom to the reaction-diffusion equations alter the simulation results?

Minor

- 1) There are many times referred to throughout the paper (15 h, 25 h, 50 h, etc.), but it is not clear which times count as "establishment phase" and which count as "post-establishment phase". How are these terms quantitatively defined in the context of the system examined in this paper?
- 2) In Equation 1 in the Methods, as well as Equations 0.3 and 0.7 in the SI, the terms corresponding to anaerobic growth on glucose lack a θ_g factor. Moreover, in Equation 0.13 in the SI, the term corresponding to acetate production during anaerobic growth lacks a $(1 - \theta_g)$ factor. Why is this the case?
- 3) It is only clear until the methods session that a mutant in *motA* was used for study. This information should be made clear in the main text. How does motility change the conclusion?
- 4) Figure 6: color bar should be given for D-G. The legend in line 893-896 seems to be for C, not E. If so, it is better to reorganize the legend to follow the panel order.
- 5) Figure 7: now that oxygen has complex distribution inside the colony, can the authors comment on what cell populations does GFP signal actually correspond to? It is constitutively expressed but GFP needs oxygen to mature – is GFP signal only coming from actively growing cells in oxygenated regions?
- 6) Figure 8: typo in line 929 – expansion -> expansion.
- 7) In *Pseudomonas* colony biofilm, redox active molecules such as phenazines were observed to show interesting spatial pattern. I am wondering if the authors could briefly discuss the implications of the current findings on those systems. Just curious.

Reviewer #2

(Remarks to the Author)

Agar colonies are routinely used in all microbiology labs. However, their spatial-temporal properties are quite complex and not well understood. The authors extended their previous theoretical model, and performed agent-based simulations. Their model suggested the mechanisms that drive colony expansion along the radial and vertical directions and the spatial distribution of cell death within the colony. Overall, the manuscript is well written, introduced the related works in sufficient detail, and convincingly explained the advancements made in the current work. I think this work could significantly advance our understanding of the development and metabolism of agar colonies, and their approach could help us better understand bacterial colonies in more natural settings. My major concern is that most of the claims are theoretical and lack experimental validation.

- 1) The model shows that there is a monolayer of cells at the edge of the colony, and the width of that layer determines the rate of colony expansion along the radial direction. However, the authors didn't provide experimental evidence to show the existence of the monolayer. The model also suggests that, since the width of the monolayer is independent of glucose concentration, the colony expansion rate is also independent of glucose concentration. It's important to experimentally validate that the monolayer width is indeed independent of glucose concentration.
- 2) The model shows that most of the cells in the colony are aligned toward the z rather than the x direction, and this configuration drives the vertical expansion of the colony. Experimental validation of the cell orientation should be provided. It should be straight forward to do with high resolution microscopy.
- 3) The authors reasoned that pH should be neutral throughout the colony (lines 428-430). However, this is highly speculative. Since pH could affect bacterial growth, they should provide direct measurements of pH within the colony and in the agar. This can be done using fluorescent dyes or sensors. If pH is not neutral, its effects should be discussed in the model.
- 4) To make the simulation computationally manageable, the authors employed a two-dimensional model. Therefore, in the model, all the cells lie within the x-z plane. However, in reality, the colonies are three-dimensional, and the cells can also tilt toward the y axis. How does the latter affect the predictions of the model?
- 5) The shape of the colony from the simulation (Fig. 3A & B) looks different from that from the experiment (Fig. 1A & B). What caused the difference?
- 6) There seems to be some discrepancy between the predicted cell death within the colony and the experimental result. Specifically, the predicted death zone shown in Fig. 7E has a flat bottom, which is very close to the agar; however, the experimental result shown in Fig. 7H revealed a death zone with a concaved-up bottom, and the non-dead zone above the agar is quite thick. What caused the difference?
- 7) The authors suggested that their findings may apply to all biofilms (line 437). This is exaggerated. Biofilms are found in a broad range of conditions: solid-air interface, solid-liquid interface, liquid-air interface, floating aggregates, host interface. Different conditions would result in different cell alignments and nutrient/oxygen distributions. The main findings in this work only apply to biofilms at solid-air interface.

Reviewer #3

(Remarks to the Author)

This paper is a follow-up to the author's paper "Spatiotemporal establishment of dense bacterial colonies growing on hard agar" (Warren et al. 2019). There, the authors focused on the early phase of colony expansion (establishments phase) and showed that mechanical forces, and not nutrient depletion, limit the radial expansion velocity, as radial movement is limited to a thin ring of fixed width. Instead, nutrient depletion only limits the vertical expansion velocity.

In this paper, the authors extend their model to study the post-establishment phase by including metabolic processes such as the finite glucose supply, oxygen limitation, production and consumption of fermentation products, and cell maintenance costs. They also introduced computational tricks to allow for longer simulations including higher numbers of cells, within reasonable time.

The authors show that the conclusions of their previous work hold also in this later phase of colony expansion: the radial velocity remains limited by mechanical forces, and not nutrient limitation, while the vertical velocity remains limited by nutrient depletion. Moreover the authors show with both their model and experimental data that there is cell death in the interior of the colony due to glucose and oxygen limitation, while at the top of the colony cells are still viable, though not growing, due to aerobic acetate consumption.

I find the paper very interesting because it relates macroscopic colony features (expansion velocities and colony shape) to single cell responses to local gradients. I appreciate the way in which the authors combined multiscale modeling with experiments. The paper is relevant because, expanding on Warren et al. 2019, it clarifies the misconception that linear radial expansion is due to nutrient depletion in the colony interior. Moreover, to my knowledge this is the first study that quantitatively links nutrient starvation to cell death in the colony center and that shows the important role of fermentation products in maintaining cell viability in the upper colony layers. I believe the paper will be of interest to a wide audience: the authors findings will be of relevance to researchers working on colonies and biofilms, moreover the multiscale modeling approach is also of interest to computational and quantitative biologists more generally.

The paper is overall well done, the methods are clearly explained and the conclusions generally well supported. However, some background info is missing, and insufficient details regarding statistical treatment are provided. Specifically, I have the following comments:

Major comments

1) It's quite hard to fully understand the novelty of the work done by the authors without going back to the author's previous work (Warren et al, 2019) as well as the work done by other people. To make this manuscript stand by itself, I believe it is essential to add a paragraph to the intro which describes the previous dogma (why linear and not exponential expansion, and why linear expansion was thought to be the result of nutrient depletion) in some more detail. Moreover, it would be important to summarize the author's previous findings (Warren et al, 2019) in some more detail, and specifically give an intuitive understanding of how mechanical factors limit the expansion speed.

2) I could not really find information regarding the number of replicates used and statistical analysis done by the authors. For most figures it is not clear if the data comes from a single replicate (and if so, how the authors checked that results are reproducible), or if it was averaged over multiple replicates (and if so, how this averaging was done). It is essential that the authors provide much more detail in the captions and in the method sections regarding these points. Moreover, it is essential that all major results are supported by data coming from multiple independent replicates, something which is impossible to judge at the moment given the lack of details.

3) It was not fully clear to me how the 1+1 ($r+z$) dimensional treatment was implemented, and how artifacts were avoided. Specifically, diffusion of metabolites into the ϕ -direction, and friction with neighbors in the ϕ -direction could both have important effects. Did the authors correct for this?

4) Finally, neither the model code nor the experimental data has been made available. Both should be deposited in public repositories before the article can be accepted.

Minor comments

5) The abstract is very technical and hard to follow for those that are not fully familiar with the previous literature. I suggest to rewrite the abstract to make it more accessible for people outside of the field.

6) Line 169: sentence not very clear, suggest to rephrase (see also comment above regarding missing background)

7) Line 177: "the direction of cell movement is highly correlated with cell orientation" this phrasing suggests the authors quantitatively analyzed data coming from their simulations, however I could not find where this analysis was done (the statement rather seems to be based on visual inspection of the figure), either the analysis should be added or this statement needs rephrasing.

8) Line 180: sentence not very clear, suggest to rephrase

9) Line 247; "consumption of glucose (for both growth and maintenance)" the figure label indicates that only consumption for maintenance is shown.

Version 1:

Reviewer comments:

Reviewer #1

(Remarks to the Author)

The authors have addressed all my questions/concerns. I therefore recommend the publication of the manuscript as it is. Congratulations to the authors on the solid work!

(Remarks on code availability)

Reviewer #2

(Remarks to the Author)

I appreciate the carefully crafted experiments and response. The authors have addressed all my concerns, and I recommend the manuscript for publication.

(Remarks on code availability)

Reviewer #3

(Remarks to the Author)

I greatly appreciate the extensive revisions done by the authors. The additional experiments, analysis, and clarifications have further strengthened the solidity of this work and have fully addressed all my concerns. I therefore recommend publishing this manuscript in its current form.

I did notice one minor typo though in line 363 (missing "the" before colony).

(Remarks on code availability)

Code files and readme are provided, I did not have a chance to test the code though.

Reviewer #4

(Remarks to the Author)

(Remarks on code availability)

**Reviewer 1**

*In this manuscript, Kannan et al. used a combination of agent-based simulations and time-lapse imaging to paint*
*a comprehensive picture of how bacterial colony grows on hard agar substrate. Colony growth is an old topic;*
*however, the authors have presented convincing evidence that the mechanism underlying the colony growth*
*dynamics differs from what researchers commonly believe in the field. While it is commonly thought that a narrow*
*growth zone at the colony edge arises due to nutrient limitations and causes linear colony expansion, the author*
*proposed, instead, that it is mechanical in origin. They further demonstrated that inside the growing colony, oxygen*
*and glucose limitation leads to complex spatiotemporal dynamics. Overall, the ideas in the manuscript are*
*interesting, and the simulation model is comprehensive. The paper could be further strengthened by a few*
*additional experiments that test their conclusions, particularly those regarding the key simulation parameters*
*underlying particular trends. To sum up, I recommend publication of the manuscript after addressing following*
*concerns/questions.*

**Response:** We are glad that the reviewer finds the ideas presented in this paper interesting. We followed up on
the reviewer's suggestions and performed new experiments to validate our conclusions and have also done
additional simulations to further strengthen the findings reported in this manuscript (described below in our
responses to the reviewer's specific comments).

**Major comments**

*1) Mechanical verticalization was suggested to lead to linear colony expansion in E. coli, which is a rod-shaped*
*bacterium. But what about spherical cells that cannot undergo verticalization? If literature evidence suggests that*
*spherical cells also form colony with linearly growing radius, it will be directly contrasting the authors' hypothesis.*
*Can the authors perform similar experimental quantifications with spherical cells such as Staphylococcus aureus or*
*Staphylococcus epidermidis?*

**Response:** We thank the reviewer for this interesting question. We first remark that the essence of the linear radial
expansion model based on cell verticalization is the pressure buildup in colony interior due to cell growth and
division (1, 2). Rod-shaped bacteria which elongate along their long axis (and divide perpendicular to the long axis)
during growth verticalize in response to the mechanical instability caused by pressure build-up in the colony
interior. At the radial edge, due to lower pressure, cells which are under surface tension remain horizontal on the
agar. However, it is unclear, in the absence of detailed investigation, how spherical cells will respond to the pressure
build up in colony interior. We discussed with the journal editor about additional experiments on cocci colonies,
and it was agreed that this was beyond the scope of this work. Nevertheless, we would like to share with the
reviewer our understanding of this problem upon reflecting on the reviewer's interesting question.

REDACTED

Indeed literature (3) suggests that *S. aureus* colonies undergo linear radial expansion under certain
conditions (**Additional Figure 1.1**). Spherical cells like *S. aureus* have a complex geometry of cell division (4). For
example, *S. aureus* has been shown to divide in three perpendicular planes over three consecutive division cycles
((4, 5), **Additional Figure 1.2**). From this division scheme, a simple model giving linear radial colony expansion can
be formulated with a colony of synchronously dividing cells advancing by one cell diameter in the horizontal
direction every three divisions; see illustration in **Additional Figure 1.2**. However, it is unknown how external forces
such as surface tension, friction, cell crowding, etc. that arise during colony development affect the formation of
such division planes. Hypothetically, surface tension could cause a restriction of division planes orthogonal to the
agar surface resulting in a monolayer of cells during initial colony formation. As cell density increases due to
growth, a pressure buildup due to cell crowding in colony interior could cause division plane formation such that
daughter cells pinch-off in the vertical direction to relieve the pressure build-up. Such a situation would be
analogous to the verticalization mechanism for rod-shaped cells. Further, effects such as sliding/spreading and
surfactant production properties of *S. aureus* (6) could also affect colony expansion properties. Thus, contrary to
the reviewer's comment, we consider that colonies of spherical cells undergoing linear radial expansion does NOT
directly contrast our findings for rod-shaped bacteria. Instead, different mechanisms such as complex cell division
geometries, surfactant production, etc. could be at play in determining radial colony expansion.

REDACTED

Additional Figure 1.2. *Illustration of the geometry of cell division in S. aureus, reproduced from the PhD thesis of Helena Maria Pinto Veiga, Instituto de Tecnologia Química e Biológica, Universidade Nova de Lisboa.*

*2) The metabolic stratification phenomenon is really intriguing. Can the author change the oxygen conditions to*
*test their predictions? One of potential experiments is to include other terminal electron acceptors such as nitrate*
*in the agar plate. In this case, the electron acceptors (NO₃ instead of O₂) come from the agar plate instead of from*
*above, so the growth dynamics should be dramatically altered. See for example experiments in PMID: 23292774.*

**Response:** We thank the reviewer for the suggestion to experiment with adding nitrate to the agar plate. To
address this suggestion, we performed new experiments and tracked colony growth dynamics with and without
nitrate added to the agar plate in limiting glucose conditions. We observe that radial expansion is unaffected, while
vertical expansion is benefitted by the addition of nitrate (**SI Fig. 8**). Based on batch culture experiments, we
observed that under anaerobic conditions with nitrate provided as an electron acceptor, the growth rate of *E. coli*
was only slightly higher than anaerobic growth without nitrate (**SI Fig. 8a**). However, adding nitrate drastically
reduced the glucose uptake during anaerobic growth thereby improving the glucose-biomass yield (**SI Fig. 8b**).
Since vertical colony expansion is primarily driven by anaerobic growth (**Fig. 4**) and is limited by glucose
concentration (**Fig. 1,2**), an improvement in the anaerobic biomass yield of glucose (*biomass accumulated per mM*
*of glucose consumed*) explains the improved vertical colony expansion with the addition of nitrate. Also, the
independence of radial expansion on nitrate addition indicates that radial expansion is not limited by oxygen
during the first two days of colony development. Thus, these results support our model and we are grateful to the
reviewer for this suggestion.

*3) Is the colony height calibrated in the experiment? A 10x air objective was used and due to the refractive index*
*mismatch between air and water/biofilm/agar, the information in z could be significantly distorted. This effect is*
*exacerbated as one goes deeper into the sample.*

**Response:** The colonies were imaged in an inverted configuration with the air objective directly facing the colony
i.e., the colony boundary and the objective was separated only by air. To determine the outline and dimensions of
a colony only its boundaries (i.e., the colony-air interface) needs to be identified (there is no requirement to go
deeper into the sample). It should be noted that similar slowdown in colony vertical dynamics have been observed
with alternate methods, using interferometry in Ref. (7) and using talcum powder in Ref. (8).

Regarding the issue of index mismatch between air and water/biofilm/agar, the only experiment which
involves probing inside the colony is the identification of the death zone. Here, the propidium iodide fluorescence
was imaged deep inside the colony using a two-photon microscope with adaptive optics. Images were obtained
using both an air objective and a water objective, both yielding similar results (**Fig. 7, SI Fig. 12-14**). For two-photon
imaging with air objective, the colonies were imaged in an upright position with the objective directly facing the
colony. For two-photon imaging with a water objective, to avoid disruption of colony surface by water, a thin layer

of low melt agarose was carefully poured on top of the colony at < 40 degree Celsius, and the colony was imaged
in an upright position with the thin agarose layer facing the water objective (described in Methods).

4) The experiments and simulations in the paper seem to have been performed only with two initial glucose
concentrations (10 mM and 20 mM). Adding, say, one or two more glucose concentrations and confirming that (a)
the simulations recapitulate the colony growth kinetics, as in Figs. 1A-B vs. 3A-B, and (b) the two main results in
the paper still hold would significantly strengthen their persuasiveness of the paper.

**Response:** Following the reviewer's suggestion, we performed additional experiments and simulations with 30
mM and 40 mM initial glucose concentration. Our findings further support the original observation that radial
expansion is similar (**Additional Figure 1.3**) during the first two days, while vertical expansion remains limited by
glucose throughout the extended range of glucose concentration (**Additional Figure 1.3**).

Additional Figure 1.3 (Experiments). The radius (μm), height (μm) and volume (μm^3) of the colonies grown in glucose minimal media agar plates (1.5% agar) are plotted against the time (h) post-inoculation for different initial glucose concentrations (10 mM, 20 mM, 30 mM, and 40 mM).

Due to a high computational load and memory constraints on the high-performance cluster we were
unable to do perform long term simulation beyond 40 h for the 40 mM initial concentration simulations (pentagons
in **Additional Figure 1.4**). Thus, we exclude the 40 mM glucose concentration experiments and simulations from
the manuscript and amend our **Fig. 1** and **Fig. 2** in the main text to additionally include only the 30 mM glucose
concentration results.

Additional Figure 1.4 (Simulations). The colony radius (μm), colony height (μm) and number of cells in the simulated colonies are plotted against the time (h) post-inoculation for different initial glucose concentrations (10 mM, 20 mM, 30 mM, and 40 mM).

*5) In the SI, the agent-based simulations and reaction-diffusion equations are presented in three dimensions, before*
*the (1+1)-dimensional approximation is presented as an implementational detail in Section 3.1. But Section 3.1*
*only discusses the agent-based simulations, not the reaction-diffusion equations, and there is no apparent*
*reference to dimensionality in Section 3.3, which concerns the solution of the reaction-diffusion equations. So in*
*what dimensions were the reaction-diffusion equations solved? Were they solved in 2D, meaning that the nutrients*
*could only diffuse in the radial and vertical directions? Or were they solved in the full 3D domain, and a vertical*
*slice of the nutrient concentrations was taken to be available to the cells being captured by the agent-based*
*simulations?*

**Response.** We thank the reviewer for this question and recognize the need for a better description of the
implementation in the main manuscript. The reaction-diffusion equations are solved in (1+1)-dimensions.
Specifically, the spatial region for these equations consists of both the rectangular agar region and the colony
region. This 2D spatial region can be viewed as a cross section (cut through the center of the colony) of the real
3D configuration of the agar-colony system. We have added additional information to clarify this in **Fig. 2e**,
**Supplementary Note 2 - section 2.2**, and also in **SI Fig. N2. 1** which illustrates the computational domain for solving
the reaction-diffusion equations.

*6) Related to the question above, the (1+1)-dimensional approximation is adequately justified, but what are the*
*potential implications of limiting the simulations (and particularly the nutrient diffusion) to two dimensions?*
*Assuming that all the cells are roughly expanding radially outward, it is perhaps reasonable to limit cell movement*
*within a vertical cross-section of the colony to only the radial and vertical directions, but this is not the case for the*
*nutrients. How might adding a degree of freedom to the reaction-diffusion equations alter the simulation results?*

**Response:** We thank the reviewer for questioning the effect of restricting the nutrient reaction-diffusion model to
a vertical cross-section. To address this, we developed a new model (described in detail in **Supplementary Note 4**)
and have performed an additional series of simulations for nutrients using two different geometries; in two
dimensions (1+1)-d, i.e., a vertical cross-section and also in three dimensions (2+1)-d, i.e., a radially symmetric
three-dimensional colony. In each simulation, we fix the size of the colony (radius and height) based on
experimental data at a certain hour during the colony growth and solve the reaction-diffusion equations with both
geometries (see **Supplementary Note 4**).

Our findings based on these new simulations indicate quantitative differences in nutrient concentration
profiles between (1+1)-d simulations and (2+1)-d simulations (**SI Fig. N4.1, N4.2** located in Supplementary Note
4). However, irrespective of the dimensions in which the reaction-diffusion system was solved, the glucose
concentration in the agar-colony interface is saturating (i.e., order of magnitude higher than the Monod constant)
for both 10 mM and 20 mM boundary condition for glucose concentration (**SI Fig. N4.1** located in Supplementary

Note 4). Further, irrespective of the dimensionality, glucose concentration drops sharply below the Monod
constant with an increasing vertical distance into the colony (SI Fig. N4.2 located in Supplementary Note 4).
Moreover, the vertical penetration of glucose into the colony is higher for 20 mM glucose condition than 10 mM
glucose condition for both geometries (SI Fig. N4.2 located in Supplementary Note 4), even though the (2+1)-d
solution has higher glucose penetration overall than the (1+1)-d solution (compare SI Fig. N4.2g and SI Fig. N4.2h
located in Supplementary Note 4). Thus, our findings based on the agent-based hybrid simulations that radial
colony expansion is not expected to be nutrient limited, while colony vertical expansion being limited by glucose
penetration remains true irrespective of the dimensions in which the reaction-diffusion equations are solved in.

**Minor comments**

*1) There are many times referred to throughout the paper (15 h, 25 h, 50 h, etc.), but it is not clear which times*
*count as "establishment phase" and which count as "post-establishment phase". How are these terms*
*quantitatively defined in the context of the system examined in this paper?*

We thank the reviewer for pointing out the lack of clarity in quantitatively defining the establishment phase and
have clarified this information in the introduction section of the revised manuscript. The establishment phase
refers to the period of colony growth between ~10h and ~20h. This was a term introduced in Warren et al. to
describe the phase where both radial and vertical colony expansion are linear in time. Any time beyond 20 h is
considered as post-establishment.

*2) In Equation 1 in the Methods, as well as Equations 0.3 and 0.7 in the SI, the terms corresponding to anaerobic*
*growth on glucose lack a θ_g factor. Moreover, in Equation 0.13 in the SI, the term corresponding to acetate*
*production during anaerobic growth lacks a $(1 - \theta_g)$ factor. Why is this the case?*

We are grateful to the reviewer for going through the equations in our model in detail. The θ_g that the reviewer
expected to appear in equation 0.3 and 0.7 (now equations 3 and 7 in Supplementary Note 2 in revised manuscript)
is included in the $\lambda_2(\lambda_{g,ana}, C_g)$ term which appears in equations 3 and 7, and is defined as: $\lambda_2(\lambda_{g,ana}, C_g) =$
$\lambda_{g,ana}\theta_g$ (in equation 4). The θ_g factor in the definition $\lambda_2(\lambda_{g,ana}, C_g) = \lambda_{g,ana}\theta_g$ serves the purpose of modeling
Monod growth law, i.e. concentration dependence of growth rate. However, the θ_g and $1 - \theta_g$ factors which
appear in equation 3 serve as “weights” to decide whether under aerobic conditions (i.e. $\theta_o = 1$) cells grow on
glucose or acetate, i.e. they model the hierarchical preference of glucose by *E. coli* for growth. In other words,
following equation 3, if glucose concentration is saturating, then $\theta_g = 1$ and $1 - \theta_g = 0$. Hence, the component
of growth rate arising from acetate is 0.

Regarding, the reviewer’s comment on lack of $1 - \theta_g$ in the term corresponding to acetate production
during anaerobic growth in equation 0.13 (now equation 13 in revised manuscript): Acetate production rate should
be proportional to the anaerobic growth rate on glucose. Anaerobic growth rate on glucose (defined in equation
3) is $\lambda_2(\lambda_{g,ana}, C_g) * (1 - \theta_o)$ where $\lambda_2(\lambda_{g,ana}, C_g) = \lambda_{g,ana}\theta_g$ (defined in equation 4). Thus, anaerobic acetate
production is $p_{a,ana}\lambda_2(\lambda_{g,ana}, C_g) * (1 - \theta_o)$ where the proportionality constant $p_{a,ana}$ denotes the specific
excretion flux of acetate during anaerobic growth on glucose (measured to be 16 mmol acetate / g_{dw} as mentioned
in Supplementary Table 1). Note that in equation 3, θ_o and $1 - \theta_o$ serve as “weights” to decide whether
metabolism is aerobic or anaerobic, i.e. if oxygen is depleted, $\theta_o = 0$; $1 - \theta_o = 1$ and thus, the only mode of
growth is anaerobic since the first and last terms in equation 3 (corresponding to aerobic growth on glucose and
acetate respectively) which have $\theta_o(=0)$ as weights are now zero. We hope that helped clarify the reviewer’s
question.

*3) It is only clear until the methods session that a mutant in motA was used for study. This information should be*
*made clear in the main text. How does motility change the conclusion?*

We have modified the manuscript to mention that we use a non-motile *E. coli* strain in the first paragraph of the
results and also in the caption of **Fig. 1** when the results on colony expansion are first presented. The role of cell
motility and chemotaxis in overall colony/range expansion in soft agar where cell swimming drives expansion has
been extensively studied in a prior manuscript by our lab (9) and others in the past (10, 11) . Regarding the effect
of motility in our study, our experiments are performed on hard agar (1.5 % w/v) where swimming and swarming
motility is not expected to play a major role. The goal of our current study is to observe colony expansion in hard
agar where cell movement and colony expansion is driven solely by cell growth and not motility, i.e. growing cells
push their neighbors to create movement and overall colony expansion. Thus, to simplify the conditions of our
study and also to make modeling the agent-based activities of cells easier we used a non-motile strain to conduct
our experiments on hard agar.

To address the reviewer’s concerns, we performed additional experiments using a motile strain which
constitutively expresses GFP to test the dependence of glucose concentration on colony growth dynamics. Radial
expansion of motile strain colonies was similar to colonies formed by non-motile strain, while the heights of motile
strain colonies were generally lower than the heights of non-motile strain colonies (**SI Fig. 15**). Flagella expression
and rotation has an energy cost and thus is expected to increase glucose consumption for the motile strain, thereby
worsening the glucose depletion inside the colony (which leads to a lower colony height). Regardless of
quantitative differences between the motile and non-motile strain colonies, our key findings on the dependence
of colony dynamics on glucose concentration remain valid (radial expansion is similar for 10 mM and 20 mM

glucose concentration, but using a higher glucose concentration increases colony height) even for colonies formed
by the motile strain (SI Fig. 15).

*4) Figure 6: color bar should be given for D-G. The legend in line 893-896 seems to be for C, not E. If so, it is better*
*to reorganize the legend to follow the panel order.*

We thank the reviewer for pointing out the lack of color bar for D-G and the error in the legend for panel C/E. We
have corrected the issues in the revised manuscript.

*5) Figure 7: now that oxygen has complex distribution inside the colony, can the authors comment on what cell*
*populations does GFP signal actually correspond to? It is constitutively expressed but GFP needs oxygen to mature*
*– is GFP signal only coming from actively growing cells in oxygenated regions?*

We thank the reviewer for raising this issue. Yes, GFP requires oxygen to mature. Indeed, as seen in **Fig. 7ij** of the
two-photon images of colonies, the GFP fluorescence is observed only near the colony-air interface and the GFP
signal sharply drops beyond a thickness $< 50 \mu\text{m}$ from the colony-air interface (**Fig. 7ij** and **SI Fig. 11cd**). In a
previous study by Díaz-Pascual et al (12), signal from a fluorescent protein (mRuby2) has been used to quantify
the oxygen gradient within the cross-section of a colony. Díaz-Pascual et al. also observe a sharp drop in mRuby2
fluorescence within $\sim 50 \mu\text{m}$ from the colony top. Further, their calibration with measurements using oxygen
microsensors suggests that when the fluorescence drop is indeed indicative of oxygen depletion. Thus, we interpret
the lack of GFP signal in colony interior as indicating a lack of GFP maturation since all cells would have synthesized
some level of GFP (due to constitutive expression) and degradation of GFP is expected to be minimal during the
duration of our study which lasts ~ 50 h (13). So, a lack of GFP signal indicates a maturation issue and we observe
no GFP signal in regions of the colony where oxygen is expected to be depleted, i.e. colony interior away from the
colony-air boundary. We have included a brief sentence on this interpretation of the GFP fluorescence signal in the
caption for **Fig. 7** of the revised manuscript.

*6) Figure 8: typo in line 929 – expansnion -> expansion.*

Thanks for catching the typo and we have corrected it in the revised manuscript.

*7) In Pseudomonas colony biofilm, redox active molecules such as phenazines were observed to show interesting*
*spatial pattern. I am wondering if the authors could briefly discuss the implications of the current findings on those*
*systems. Just curious.*

We thank the reviewer for pointing us to the role of phenazines in the context of biofilms. Pseudomonas biofilm with
phenazine presents an interesting application of our model especially in light of the spatiotemporal dynamics of
oxygen gradient at play within a developing colony. We are developing a model to probe the role of such redox
molecules and will report its findings in a separate study.

**References for response to reviewer 1**

- 1. M. R. Warren, *et al.*, Spatiotemporal establishment of dense bacterial colonies growing on hard agar. *eLife* **8**,
e41093 (2019).
- 2. F. Beroz, *et al.*, Verticalization of bacterial biofilms. *Nat. Phys.* **14**, 954–960 (2018).
- 3. C. Vulin, N. Leimer, M. Huemer, M. Ackermann, A. S. Zinkernagel, Prolonged bacterial lag time results in
small colony variants that represent a sub-population of persisters. *Nat. Commun.* **9**, 4074 (2018).
- 4. H. Tzagoloff, R. Novick, Geometry of cell division in *Staphylococcus aureus*. *J. Bacteriol.* **129**, 343–350 (1977).
- 5. H. Veiga, A. M. Jorge, M. G. Pinho, Absence of nucleoid occlusion effector Noc impairs formation of
orthogonal FtsZ rings during *Staphylococcus aureus* cell division. *Mol. Microbiol.* **80**, 1366–1380 (2011).
- 6. E. J. G. Pollitt, S. P. Diggle, Defining motility in the Staphylococci. *Cell. Mol. Life Sci.* **74**, 2943–2958 (2017).
- 7. P. Bravo, S. Lung Ng, K. A. MacGillivray, B. K. Hammer, P. J. Yunker, Vertical growth dynamics of biofilms. *Proc.*
*Natl. Acad. Sci.* **120**, e2214211120 (2023).
- 8. J. W. T. Wimpenny, The Growth and Form of Bacterial Colonies. *Microbiology* **114**, 483–486 (1979).
- 9. J. Cremer, *et al.*, Chemotaxis as a navigation strategy to boost range expansion. *Nature* **575**, 658–663 (2019).
- 10. J. Adler, Chemotaxis in Bacteria. *Science* **153**, 708–716 (1966).
- 11. J. Adler, Chemoreceptors in Bacteria. *Science* **166**, 1588–1597 (1969).
- 12. F. Díaz-Pascual, *et al.*, Spatial alanine metabolism determines local growth dynamics of *Escherichia coli*
colonies. *eLife* **10**, e70794 (2021).
- 13. T. Gervais, *et al.*, *E. coli* prepares for starvation by dramatically remodeling its proteome in the first hours
after loss of nutrients. [Preprint] (2025). Available at:
<https://www.biorxiv.org/content/10.1101/2024.02.29.582700v2> [Accessed 22 January 2025].

**Reviewer 2**

*Agar colonies are routinely used in all microbiology labs. However, their spatial-temporal properties are quite*
*complex and not well understood. The authors extended their previous theoretical model, and performed agent-*
*based simulations. Their model suggested the mechanisms that drive colony expansion along the radial and vertical*
*directions and the spatial distribution of cell death within the colony. Overall, the manuscript is well written,*
*introduced the related works in sufficient detail, and convincingly explained the advancements made in the current*
*work. I think this work could significantly advance our understanding of the development and metabolism of agar*
*colonies, and their approach could help us better understand bacterial colonies in more natural settings. My major*
*concern is that most of the claims are theoretical and lack experimental validation.*

**Response:** We thank the reviewer for recognizing the value of our approach to better understand bacterial colony
growth. To address the reviewer's concern for lack of experimental validation we have performed several new
experiments and additional modelling/simulations (described in our responses to the reviewer's specific
comments below). Overall, we find that the new experiments are in agreement with our model predictions
(described below in our responses).

*1) The model shows that there is a monolayer of cells at the edge of the colony, and the width of that layer*
*determines the rate of colony expansion along the radial direction. However, the authors didn't provide*
*experimental evidence to show the existence of the monolayer. The model also suggests that, since the width of*
*the monolayer is independent of glucose concentration, the colony expansion rate is also independent of glucose*
*concentration. It's important to experimentally validate that the monolayer width is indeed independent of glucose*
*concentration.*

**Response:** In our experiments, colonies that originate as a single cell initially expand as a single layer of cells, and
then verticalize to form a multilayered 3D colony upon reaching a critical radius (**SI Fig. 2jk**). The establishment of
a monolayer of cells at the radial edge of colonies during this initial period of colony growth was previously shown
by Su et al. 2012 (1) and Warren et al. 2019 (2). Su et al found verticalization to happen at ~ 4h and observed a
monolayer width of ~20 μm maintained up to the duration of their observation (~ 8h) (reproduced in **Additional**
**Figure 2.1.** below from Ref. (1)).

To address the reviewer's comment and to experimentally track the monolayer at later stages of colony
development, we performed additional experiments. The peripheral region of the colonies were imaged using high
resolution confocal microscopy (see Methods). It is indeed seen that there is a ~20 μm region at the radial edge
comprised of a monolayer of cells (**Supplementary Videos 3-4, SI Fig. 5**) that remains roughly constant throughout
development till 50 h (**Fig. 3m**). Further, the width of this monolayer region is similar for colonies grown with both

10 mM and 20 mM glucose minimal media agar plates (**Fig. 3klm**). It should be noted that the quantitative
differences in monolayer width predicted by our simulations ($\sim 60 \mu\text{m}$) and experiments ($\sim 20 \mu\text{m}$) is likely due to
the (1+1)-dimensional geometry of our simulations wherein the cells in the monolayer are forced to align their
long axis in the radially outward direction which is not necessarily the case in experiments (see **Fig. 3a** for
simulation and **Fig. 3kl, SI Fig. 5** for experiments). Such a (1+1)-geometry is thus indeed expected to overestimate
the monolayer width. Further, such width is likely to depend on model parameters such as friction coefficients,
surface tension parameters, etc.

*2) The model shows that most of the cells in the colony are aligned toward the z rather than the x direction, and*
*this configuration drives the vertical expansion of the colony. Experimental validation of the cell orientation should*
*be provided. It should be straight forward to do with high resolution microscopy.*

**Response:** Vertical orientation of cells within the interior of bacterial biofilms has been extensively characterized
in *Pseudomonas aeruginosa* biofilms (3) and *Vibrio cholerae* biofilms (4). In particular, Dayton et al. by using
scanning electron microscopy (SEM) of vertical cross sections of mature colonies showed verticalization of
*Pseudomonas* cells and their alignment towards the z direction in the colony interior. Motivated by Dayton et al.

and the reviewer comment, we performed additional experiments and repeated protocol described in Ref. (3) for
sample preparation and SEM for our *E. coli* colonies. We indeed find high orientational order of cells towards the
vertical direction in the colony interior (**SI Fig. 4**).

*3) The authors reasoned that pH should be neutral throughout the colony (lines 428-430). However, this is highly*
*speculative. Since pH could affect bacterial growth, they should provide direct measurements of pH within the*
*colony and in the agar. This can be done using fluorescent dyes or sensors. If pH is not neutral, its effects should be*
*discussed in the model.*

**Response:** We thank the reviewer for bringing up this important point and recognize the need for additional
experiments and a more explicit reasoning than the text presented in the previous draft of the manuscript. The
effects of acid stress on *E. coli* growth has been extensively studied in a prior manuscript from our lab (5). It is
known that *E. coli* growth rate is roughly constant across the pH range of 5-7 if the pH drop is due to inorganic
acids such as HCl (5) (**Additional Figure 2.2a**). However, growth is affected by the accumulation of short chain fatty
acids (SCFA) such as acetic acid in the pH range of 5-7 (5) (**Additional figure 2.2bc**). Since acetate is excreted by
cells in our colony and acetate can both lower the pH and is an agent of SCFA stress, measuring pH alone would
not be sufficient to answer the reviewer's concerns. Instead, we performed additional experiments to probe SCFA
stress as described below. These experiments allow us to address the reviewer's question in a more quantitative
way than using dyes to measure pH inside a dense 200 μm thick colony where light cannot penetrate well.

REDACTED

The effect of acetate on pH depends crucially on the buffer capacity of the medium. Roughly, in a medium
with buffer concentration X , acetate at concentration $\ll X$ would have little effect on the pH (see Supplementary
Note 5 for detailed calculations). In the experiments reported in our manuscript, the initial pH of the agar medium

was set to 7.2 with a phosphate buffer at a very high concentration of 112 mM. If cells in our 112 mM buffer
conditions were under SCFA stress, then using a lower buffer concentration (which would further reduce pH and
exacerbate SCFA stress) would result in a further reduction in cell growth. However, using a 33 mM buffer had
minimal effect in the overall colony growth dynamics (**SI Fig. 9**). This can be rationalized as follows: Excretion of
SCFA (primarily acetate (5)) by cells growing on glucose within the colony could potentially cause acid stress. Since
the only external carbon source provided is glucose, at 20 mM glucose, an upper bound for acetate concentration
is ~ 45 mM (based on anaerobic glucose metabolism ~2.3 mM of SCFA is excreted per 1 mM of glucose consumed
(5)). Note that 45mM is a gross over-estimate for acetate concentration since glucose consumption within the
colony is a combination of aerobic and anaerobic metabolism and acetate excretion is much lower during aerobic
glucose metabolism. Indeed, our simulations show that the local acetate concentration level inside the colony is
no more than ~ 8 mM (**Fig. 5**). At these low acetate levels, the pH will remain near-neutral even with 33 mM buffer,
and hence the impact on growth rate of cells is expected to be minimal (detailed calculations described in
**Supplementary Note 5**).

However, using an 11 mM buffer is expected to severely affect cell growth even for low acetate
concentration <10 mM (**Supplementary Note 5**). Indeed, the colony height is drastically reduced when an 11 mM
buffer is used (**Additional Figure 2.3**). Further, the colonies lost radial symmetry and have irregular expanding
fronts that are highly variable across colonies (**Additional Figure 2.3**). Also, in **Additional Figure 2.3. i-k**, it is seen
that the pH near the colonies progressively acidifies as the colony grows in an 11 mM buffer plate (indicated by
the yellow color of the bromocresol purple dye mixed into the agar). Note that this dye remains purple throughout
colony development for 112 mM buffer plates (**Additional Figure 2.3. f-h**).

Thus, our data shows that in our culture conditions of 112 mM buffer, pH is not expected to play a
significant role. We have added a paragraph describing this matter in the revised manuscript. Further, we would
also like to remark that the effect of pH and SCFA stress on colony development is the topic of an ongoing study
by us, but beyond the scope intended for this manuscript.

**Additional Figure 2.3. Colony expansion dynamics is severely impacted at 11 mM buffer.** Expansion dynamics of EQ59
 *E. coli* colonies on 1.5 % (w/v) agar plates prepared with 20 mM glucose, 10 mM ammonium chloride and a defined
 concentration (11 mM and 112 mM) of phosphate buffer) at various times post-inoculation as a single cell. **(a)** The height
 of colonies grown with 11 mM buffer and 112 mM buffer. **(b)** Radius of the bounding circle for colonies grown in 11 mM
 buffer. Each type of symbol represents a different biological replicate. **(c)** Photograph of a minimal media 11 mM buffer
 agar plate with colonies. The biological variability in radial dimension for 11 mM buffer condition is apparent in (b, c).
 Comparison of cross-section profile of a colony grown with 112 mM buffer and 11 mM buffer at 13h **(d)**, and 40h **(e)**.
 (a,b,d,e) Cyan symbols represent colonies grown on minimal media plates with 112 mM buffer while magenta symbols
 represent colonies with 11 mM buffer. Minimal media agar plates containing bromocresol purple with colonies grown
 with 112 mM buffer **(f-h)**, and 11 mM buffer **(i-k)** at various times of development. Purple color indicates a neutral pH
 while yellow indicates acidification. Visually, it is seen that the peripheral regions near the boundaries of colonies with
 11 mM buffer progressively becomes yellow over time indication acidification **(i-k)** unlike colonies with 112 mM buffer
 **(f-h)**.

*4) To make the simulation computationally manageable, the authors employed a two-dimensional model.*
*Therefore, in the model, all the cells lie within the x-z plane. However, in reality, the colonies are three-dimensional,*
*and the cells can also tilt toward the y axis. How does the latter affect the predictions of the model?*

**Response:** Indeed, cells can also tilt toward the y axis in real colonies unlike in our (1+1)-agent based model. As
discussed in our response to the reviewer's comment 1, this directly causes a quantitative difference in the
monolayer width in our simulations and experiments. Cells in the monolayer of our simulations are forced to point
their long axis radially outwards, unlike in experiments which lead to an overestimation of the monolayer width in
our simulations. In regards to the effect on vertical expansion, based on prior studies on cell verticalization in
colonies (3, 4) and our SEM images of colony cross section (**SI Fig. 4**), there is strong orientational order of cells
toward the z-axis in the colony with minor tilt in the y-direction. Thus, growth of these interior cells is expected to
predominantly contribute to vertical colony expansion along the z-direction. Any minor tilt of cells along the y-axis
would contribute to less colony expansion along the z-direction and correspondingly more towards the y-direction.
Colony expansion in the y-direction would thereby cause an overall flattening/rounding effect to the morphology
of the colony as opposed to the very triangular shaped colonies arising in our agent-based simulations within a
(1+1)-dimensional geometry.

*5) The shape of the colony from the simulation (Fig. 3A & B) looks different from that from the experiment (Fig. 1A*
*& B). What caused the difference?*

**Response:** As the reviewer astutely foreshadowed in their previous comment, we believe that the difference in
the shape of the simulated colony and the experiment may be attributed to the orientation of cells being restricted
to the x-z plane, i.e. the vertical cross-section in our agent-based model. Any minor tilt of cells along the y-axis
would be expected to contribute to more growth and colony expansion along the y-direction. This would result in
a more rounded colony shape like observed in experiments (**Fig. 1abc**) as opposed to the very triangular shaped
colonies arising in our agent-based simulations restricted to a (1+1)-dimensional geometry (**Fig. 2ghi**).

*6) There seems to be some discrepancy between the predicted cell death within the colony and the experimental*
*result. Specifically, the predicted death zone shown in Fig. 7E has a flat bottom, which is very close to the agar;*
*however, the experimental result shown in Fig. 7H revealed a death zone with a concaved-up bottom, and the non-*
*dead zone above the agar is quite thick. What caused the difference?*

**Response:** We remark that **Fig. 7e** reports a probability of death based on our model combining agent-based
simulations and measurements of batch culture death rates. The experimental result reported in Fig. 7H (**Fig. 7j** in
revised draft) is fluorescence signal from propidium iodide (PI) which fluoresces upon binding to nuclei acids and
is cell membrane impermeant. Thus, the PI fluorescence reports the local nuclei acid concentration arising from

both membrane-compromised cells and also any potential extracellular nuclei acids leaking from such dead cells.
Note that PI fluorescence is reporting a downstream consequence of death rather than the location of dead cells
directly. Thus, quantitatively relating the death probability predicted from the model and PI fluorescence would
depend on the relationship between the number of dead cells in a voxel and the PI fluorescence intensity which is
difficult to calibrate in vivo inside a dense colony. Our interpretation of the localization of extracellular nuclei acids
in colony interior is to qualitatively attribute it to the increased presence of dead cells, i.e., cells with compromised
membranes in the colony interior which is consistent with our model prediction. However, extracellular nucleic
acids will be consumed by neighboring alive cells creating a reaction-diffusion system by itself and thereby
generating a spatial gradient of nuclei acid concentration (PI fluorescence). Thus, relating the shape and concavity
of the nuclei acid concentration to the precise location of dead cells requires highly non-trivial modeling and also
additional details on the physiology of such carcass-refeeding / cannibalism. We would like to mention that the
consequences of such internal cross-feeding within the colony are the topic of an ongoing study by us, but beyond
the scope intended for this manuscript.

*7) The authors suggested that their findings may apply to all biofilms (line 437). This is exaggerated. Biofilms are*
*found in a broad range of conditions: solid-air interface, solid-liquid interface, liquid-air interface, floating*
*aggregates, host interface. Different conditions would result in different cell alignments and nutrient/oxygen*
*distributions. The main findings in this work only apply to biofilms at solid-air interface.*

**Response:** We agree with the reviewer's comment and removed that sentence in the revised manuscript.

**References for response to reviewer 2**

- 1. P.-T. Su, *et al.*, Bacterial Colony from Two-Dimensional Division to Three-Dimensional Development. *PLOS*
*ONE* **7**, e48098 (2012).
- 2. M. R. Warren, *et al.*, Spatiotemporal establishment of dense bacterial colonies growing on hard agar. *eLife* **8**,
e41093 (2019).
- 3. H. Dayton, *et al.*, Cellular arrangement impacts metabolic activity and antibiotic tolerance in *Pseudomonas*
*aeruginosa* biofilms. *PLOS Biol.* **22**, e3002205 (2024).
- 4. F. Beroz, *et al.*, Verticalization of bacterial biofilms. *Nat. Phys.* **14**, 954–960 (2018).
- 5. B. R. Taylor, *et al.*, A metabolic sum rule dictates bacterial response to short-chain fatty acid stress. [Preprint]
(2022). Available at: <https://www.biorxiv.org/content/10.1101/2022.08.31.506075v1> [Accessed 8 July 2023].

**Reviewer 3**

*This paper is a follow-up to the author's paper "Spatiotemporal establishment of dense bacterial colonies growing*
*on hard agar" (Warren et al. 2019). There, the authors focused on the early phase of colony expansion*
*(establishments phase) and showed that mechanical forces, and not nutrient depletion, limit the radial expansion*
*velocity, as radial movement is limited to a thin ring of fixed width. Instead, nutrient depletion only limits the*
*vertical expansion velocity.*

*In this paper, the authors extend their model to study the post-establishment phase by including metabolic*
*processes such as the finite glucose supply, oxygen limitation, production and consumption of fermentation*
*products, and cell maintenance costs. They also introduced computational tricks to allow for longer simulations*
*including higher numbers of cells, within reasonable time.*

*The authors show that the conclusions of their previous work hold also in this later phase of colony expansion: the*
*radial velocity remains limited by mechanical forces, and not nutrient limitation, while the vertical velocity remains*
*limited by nutrient depletion. Moreover the authors show with both their model and experimental data that there*
*is cell death in the interior of the colony due to glucose and oxygen limitation, while at the top of the colony cells*
*are still viable, though not growing, due to aerobic acetate consumption.*

*I find the paper very interesting because it relates macroscopic colony features (expansion velocities and colony*
*shape) to single cell responses to local gradients. I appreciate the way in which the authors combined multiscale*
*modeling with experiments. The paper is relevant because, expanding on Warren et al. 2019, it clarifies the*
*misconception that linear radial expansion is due to nutrient depletion in the colony interior. Moreover, to my*
*knowledge this is the first study that quantitatively links nutrient starvation to cell death in the colony center and*
*that shows the important role of fermentation products in maintaining cell viability in the upper colony layers. I*
*believe the paper will be of interest to a wide audience: the authors findings will be of relevance to researchers*
*working on colonies and biofilms, moreover the multiscale modeling approach is also of interest to computational*
*and quantitative biologists more generally.*

*The paper is overall well done, the methods are clearly explained and the conclusions generally well supported.*
*However, some background info is missing, and insufficient details regarding statistical treatment are provided.*

**Response:** We are glad that the reviewer finds our results and the multi-scale modeling approach in our manuscript
interesting. We also appreciate the reviewer recognizing the aspect of our study which links nutrient starvation to
the development of a death-zone in the colony interior. To address the reviewer's concerns of missing background
information, we have significantly rewritten our manuscript to relate our study and existing literature. Further, we

have also clarified details regarding the statistical treatment, i.e. the number of biological replicates for
experiments in the figures, figure captions and also in the source data provided for our experiments.

*Specifically, I have the following comments:*

**Major comments**

*1) It's quite hard to fully understand the novelty of the work done by the authors without going back to the author's*
*previous work (Warren et al, 2019) as well as the work done by other people. To make this manuscript stand by*
*itself, I believe it is essential to add a paragraph to the intro which describes the previous dogma (why linear and*
*not exponential expansion, and why linear expansion was thought to be the result of nutrient depletion) in some*
*more detail. Moreover, it would be important to summarize the author's previous findings (Warren et al, 2019) in*
*some more detail, and specifically give an intuitive understanding of how mechanical factors limit the expansion*
*speed.*

**Response:** We thank the reviewer for pointing this out and understand it is important to provide sufficient
information in the manuscript to guide the readers through our findings. To this end, we significantly reworked the
introduction. In the second paragraph of the introduction, we describe the canonical understanding of colony
growth, first put forth by Pirt in 1967 (1) and also mention the subsequent challenges of this model by Wimpenny
(2). Further, we provide a summary of the contribution by Warren et al in 2019 (3) towards a quantitative
understanding of radial and vertical colony expansion, as well as the limitations of that study.

*2) I could not really find information regarding the number of replicates used and statistical analysis done by the*
*authors. For most figures it is not clear if the data comes from a single replicate (and if so, how the authors checked*
*that results are reproducible), or if it was averaged over multiple replicates (and if so, how this averaging was*
*done). It is essential that the authors provide much more detail in the captions and in the method sections regarding*
*these points. Moreover, it is essential that all major results are supported by data coming from multiple*
*independent replicates, something which is impossible to judge at the moment given the lack of details.*

**Response:** All of our experimental results for colony dynamics are from multiple biological replicates. We have
revised our figures to clearly represent replicate information with each independent replicate marked by a
different symbol in the figures.

*3) It was not fully clear to me how the 1+1 (r+z) dimensional treatment was implemented, and how artifacts were*
*avoided. Specifically, diffusion of metabolites into the phi-direction, and friction with neighbors in the phi-direction*
*could both have important effects. Did the authors correct for this?*

**Response:**

We have made several modifications to our manuscript to clarify the implementation of the (1+1)-dimensional
treatment. Briefly, the reaction-diffusion equations are solved in (1+1)-dimensions. Specifically, the spatial region
for these equations consists of both the rectangular agar region and the colony region. We have added additional
information to clarify this in **Supplementary Note 2 - section 2.2** and also in **SI Fig. N2.1** which illustrates the
computational domain for solving the reaction-diffusion equations. We have also included explicit indications that
the reaction-diffusion equations are solved in (1+1)-dimensions in the illustrations **Fig. 2ef** of main text. Further,
for the agent-based component of model, the cells are confined to the (1+1)-dimensional geometry by artificially
setting the out-of-plane component of forces and cell movement to zero (**Supplementary Note 3**).

We also thank the reviewer for questioning the effect of such (1+1)-dimensional treatment on our results.
To specifically address the effect of reduced spatial dimension on metabolite diffusion, we developed a new model
(described in detail in **Supplementary Note 4**) and have performed an additional series of simulations for nutrients
using two different geometries; in (1+1)-d, i.e., a vertical cross-section and also in (2+1)-d, i.e., a radially symmetric
three-dimensional colony. In each simulation, we fix the size of the colony (radius and height) based on
experimental data at a certain hour during the colony growth and solve the reaction-diffusion equations with both
geometries (see **Supplementary Note 4**). Our findings based on these new simulations indicate quantitative
differences in nutrient concentration profiles between (1+1)-d simulations and (2+1)-d simulations (**SI Fig. N4.1,**
**N4.2** located in Supplementary Note 4). However, irrespective of the dimensions in which the reaction-diffusion
system was solved, the glucose concentration in the agar-colony interface is saturating (i.e., order of magnitude
higher than the Monod constant) for both 10 mM and 20 mM boundary condition for glucose concentration (**SI**
**Fig. N4.1** located in Supplementary Note 4). Further, irrespective of the dimensionality, glucose concentration
drops sharply below the Monod constant with an increasing vertical distance into the colony (**SI Fig. N4.2** located
in Supplementary Note 4). Moreover, the vertical penetration of glucose into the colony is higher for 20 mM
glucose condition than 10 mM glucose condition for both geometries (**SI Fig. N4.2** located in Supplementary Note
4), even though the (2+1)-d solution has higher glucose penetration overall than the (1+1)-d solution (compare **SI**
**Fig. N4.2g** and **SI Fig. N4.2h** located in Supplementary Note 4). Thus, our findings based on the agent-based hybrid
simulations that radial colony expansion is not expected to be nutrient limited, while colony vertical expansion
being limited by glucose penetration remains true irrespective of the dimensions in which the reaction-diffusion
equations are solved in.

Regarding the effect of reduced spatial dimension on friction with neighbors and also other mechanical
aspects, there are several potential consequences due to the restriction of geometry to the x-z plane. Particularly
in the monolayer region at colony periphery, cells are forced to point their long axis radially outwards (**Fig. 3a**),
unlike in experiments (**Fig. 3kl, SI Fig. 5 and Supplementary Videos 3,4**). This leads to an incomplete description
of cell orientation in the monolayer and also an overestimation of the monolayer width in our simulations. Further,
in the interior region of simulated colonies, cells are forced to not have any tilt in the out-of-plane i.e., y-direction.
However, in experimental colonies, cells may have a minor tilt along y-axis in the colony interior and the resulting
friction, elastic repulsion caused by neighbors in the y-direction would contribute to colony expansion along the y-
direction. This could potentially explain the discrepancy in morphology between simulated and experimental
colonies, i.e. experimental colonies have a more rounded colony shape (due to expansion along y-direction) (**Fig.**
**1abc**) as opposed to the very pointy, triangular shaped colonies arising in our agent-based simulations restricted
to a (1+1)-dimensional x-z geometry (**Fig. 2ghi**). However, such potential quantitative and morphological effects
arising from reduced spatial dimension are not expected to impact the major findings we are focusing on in this
work, i.e. radial colony expansion is not nutrient limited, while colony vertical expansion is limited by glucose
penetration with a necrotic death zone in the colony interior. We have added a discussion of such issues in
Supplementary Note 4 of our revised manuscript.

*4) Finally, neither the model code nor the experimental data has been made available. Both should be deposited in*
*public repositories before the article can be accepted.*

**Response:** We have uploaded the code on a publicly accessible repository on GitHub and have also provided the
source data for figures to the journal.

**Minor comments**

*5) The abstract is very technical and hard to follow for those that are not fully familiar with the previous literature.*
*I suggest to rewrite the abstract to make it more accessible for people outside of the field.*

We have rewritten the abstract by replacing some of the field-specific terms such as cell differentiation, etc. We
hope that the revised abstract is more accessible to the general audience now.

*6) Line 169: sentence not very clear, suggest to rephrase (see also comment above regarding missing background)*

*7) Line 177: "the direction of cell movement is highly correlated with cell orientation" this phrasing suggests the*
*authors quantitatively analyzed data coming from their simulations, however I could not find where this analysis*

*was done (the statement rather seems to be based on visual inspection of the figure), either the analysis should be*
*added or this statement needs rephrasing.*

*8) Line 180: sentence not very clear, suggest to rephrase*

Regarding reviewer comments 6, 7 & 8: we have completely rewritten the paragraph containing the mentioned
Lines 169, 177 & 180 in the revised manuscript to improve clarity.

*9) Line 247; “consumption of glucose (for both growth and maintenance)” the figure label indicates that only*
*consumption for maintenance is shown.*

We have removed the above line (and the mentioned figure) from the revised manuscript as we have significantly
rewritten the paragraph containing line 247 to focus on the results presented in the new Figure 5 which discusses
the role of acetate consumption in colony growth.

**References for response to reviewer 3**

- 1. S. J. Pirt, A Kinetic Study of the Mode of Growth of Surface Colonies of Bacteria and Fungi. *Microbiology* **47**,
181–197 (1967).
- 2. J. W. T. Wimpenny, The Growth and Form of Bacterial Colonies. *Microbiology* **114**, 483–486 (1979).
- 3. M. R. Warren, *et al.*, Spatiotemporal establishment of dense bacterial colonies growing on hard agar. *eLife* **8**,
e41093 (2019).

**REVIEWERS' COMMENTS – FINAL REVISIONS**

**Reviewer #1 (Remarks to the Author):**

*The authors have addressed all my questions/concerns. I therefore recommend the publication of the manuscript*
*as it is. Congratulations to the authors on the solid work!*

We thank the reviewer for their insightful comments that improved the solidity of the work and their
recommendation for publication.

**Reviewer #2 (Remarks to the Author):**

*I appreciate the carefully crafted experiments and response. The authors have addressed all my concerns, and I*
*recommend the manuscript for publication.*

We thank the reviewer for their insightful review of our work and their recommendation for publication.

**Reviewer #3 (Remarks to the Author):**

*I greatly appreciate the extensive revisions done by the authors. The additional experiments, analysis, and*
*clarifications have further strengthened the solidity of this work and have fully addressed all my concerns. I*
*therefore recommend publishing this manuscript in its current form.*

We thank the reviewer for their thoughtful comments on our work and their recommendation for publication.

*I did notice one minor typo though in line 363 (missing “the” before colony).*

The typo has been fixed in the revised manuscript.

**Reviewer #3 (Remarks on code availability):**

*Code files and readme are provided, I did not have a chance to test the code though.*

We have deposited the code to a publicly accessible online repository with a demo example to reduce the
barrier to test for future users.